# Spatio-temporal variations of water sources and mixing spots in a riparian zone

Guilherme E. H. Nogueira[1]; Christian Schmidt [2]; Daniel Partington [3]; Philip Brunner [4]; Jan H. Fleckenstein[1,5]

[1] Department of Hydrogeology, Helmholtz-Centre for Environmental Research - UFZ, Leipzig, Germany.
[2] Department of Aquatic Ecosystem Analysis, Helmholtz-Centre for Environmental Research - UFZ, Magdeburg, Germany.
[3] National Centre for Groundwater Research and Training, & College of Science and Engineering, Flinders University, Adelaide, Australia.
[4] Centre for Hydrogeology and Geothermics, University of Neuchâtel, Neuchâtel, Switzerland.
[5] Bayreuth Centre of Ecology and Environmental Research, University of Bayreuth, Bayreuth, Germany.

*Correspondence to*: Guilherme E. H. Nogueira (*guilherme.nogueira@ufz.de*)

**Abstract.** Riparian zones are known to modulate water quality in stream-corridors. They can act as buffers for groundwater borne solutes before they enter the stream at harmful, high concentrations, or facilitate solute turnover and attenuation in zones where stream water (SW) and groundwater (GW) mix. This natural attenuation capacity is strongly controlled by the dynamic exchange of water and solutes between the stream and the adjoining aquifer, creating potential for mixing-dependent reactions to take place. Here, we couple a previously calibrated transient and fully-integrated 3D surface-subsurface, numerical flow model with a Hydraulic Mixing Cell (HMC) method to map the source composition of water along a net losing reach (900m) of the 4th-order Selke stream and track its spatio-temporal evolution. This allows us to define zones in the aquifer with more balanced fractions of the different water sources per aquifer volume (called "mixing hot-spots"), which have a high potential to facilitate mixing-dependent reactions and in turn enhance solute turnover. We further evaluated the HMC results against hydrochemical monitoring data. Our results show that on average about 50% of the water in the alluvial aquifer consists of infiltrating SW. Within about 200m around the stream the aquifer is almost entirely made up of infiltrated SW with practically no significant amounts of other water sources mixed in. On average, about 9% of the model domain could be characterized as "mixing hot-spots", which were mainly located at the fringe of the geochemical hyporheic zone rather than below or in the immediate vicinity of the streambed. This percentage could rise to values nearly 1.5 times higher following large discharge events. Moreover, event intensity (magnitude of peak flow) was found to be more important for the increase of mixing than event duration. Our modelling results further suggest that discharge events more significantly increase mixing potential at greater distances from the stream. In contrast near and below the stream, the rapid increase of SW influx shifts the ratio between the water fractions to SW, reducing the potential for mixing and the associated reactions. With this easy-to-transfer framework we seek to show the applicability of the HMC method as a complementary approach for the identification of mixing hot-spots in stream corridors, while showing the spatio-temporal controls of the SW-GW mixing process and the implications for riparian biogeochemistry and mixing-dependent turnover processes.

# 1 Introduction

## 1.1 Importance of mixing at the riparian zone

The importance of riparian zones for regulating water quality in stream corridors has long been recognized (Bernhardt et al., 2017; Gu et al., 2012; Hill, 1996; Jencso et al., 2010; Mayer et al., 2006; McClain et al., 2003; Vidon et al., 2010). Their natural attenuation capacity is partly related to dynamic water and solute exchanges between the stream and aquifer. Specifically, the mixing of stream water (SW) and groundwater (GW) within the riparian zone increases the potential for biogeochemical reactions by bringing different reactants in contact (Gassen et al., 2017; Hester et al., 2014, 2019; Sawyer, 2015; Sawyer et al., 2014; Trauth et al., 2015). For instance, riparian zones have shown large removal capacities for nitrate ($NO_3^-$) derived from nitrogen-based fertilizers leaking into groundwater below agricultural areas (Ocampo et al., 2006; Pinay et al., 2015; Ranalli and Macalady, 2010; Vidon and Hill, 2004). Particularly along losing stream sections, infiltrating SW can increase the availability of dissolved organic carbon (DOC) as an electron donor in the riparian aquifer and in turn enhance denitrification rates following oxygen depletion (Battin, 1999; Trauth et al., 2018; Zarnetske et al., 2011).

The transit of a stream water parcel after infiltration across the streambed into the riparian aquifer is followed by progressive mixing with ambient groundwater. Here, we refer to mixing at the macroscopic level, i.e. the colocation of different source waters within a defined volume of the aquifer (e.g. a numerical model cell or element), rather than pore-scale, physical-mixing, which leads to solute molecules being present simultaneously in an overlapping area (Bear and Verruijt, 1987; Cirpka and Kitanidis, 2000; Dentz et al., 2011; Kitanidis, 1994). Increased macroscopic mixing, however, will in turn also lead to increased potential for physical mixing and associated reactions. In this sense, several studies have showed how macroscopic SW-GW mixing dynamics can control biogeochemical reactions within the riparian zone (Hester et al., 2013; McClain et al., 2003; Sawyer, 2015; Sawyer et al., 2014; Sawyer and Cardenas, 2009; Song et al., 2018; Stegen et al., 2016). For example, Hester et al. (2019) have demonstrated that increasing stream stage enhanced the mixing-dependent denitrification of upwelling $NO_3^-$, with a concomitant shift of the SW-GW mixing-interface to deeper parts of the hyporheic zone (HZ). Moreover, it has been proven that the highest potential for mixing-dependent turnover of groundwater-borne solutes is at the fringe of the HZ, where mixing between infiltrating SW and local flowing GW might develop to a larger degree (Hester et al., 2014, 2017, 2019; Sawyer and Cardenas, 2009; Trauth et al., 2015; Triska et al., 1989), Fig.1. These mixing-triggered processes could represent the last natural protection before harmful groundwater-borne solutes such as $NO_3^-$ enter a stream. As SW-GW exchange (and subsequent SW-GW mixing) is a spatially and temporally dynamic process, identifying the different water sources within the riparian zone and their mixing dynamics can be helpful to advise adequate stream restoration plans to improve aquatic ecosystem health (Hester et al., 2017; Lawrence et al., 2013). However, to the best of our knowledge, the continuous spatio-temporal changes of SW-GW mixing degrees due to transient hydrological forces have rarely been assessed at the stream corridor scale (Berezowski et al., 2019; Gomez-Velez et al., 2017; Lessels et al., 2016; Liggett et al., 2015). This is partly due to the significant effort required to identify different water sources and their dynamics at high spatio-temporal resolution at the river corridor scale.

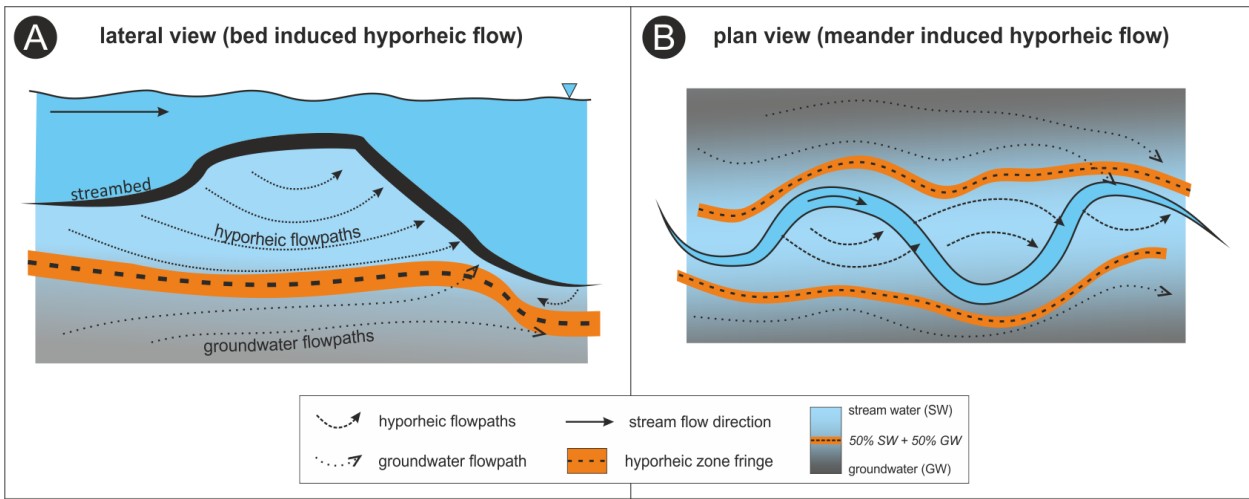

**Fig.1:** Scheme of two different hyporheic flow types and their flow paths. The orange area highlights the hyporheic zone fringe
(Triska *et al.*, 1989), with highest turnover potential for groundwater-borne solutes.

### 1.2 Identification of water sources and their relative abundance in the riparian zone.

In order to map different water sources and assess the locations and temporal variations of SW-GW exchange
processes at different scales within stream corridors, end-member mixing models (e.g., with Chloride, or other
traceable chemical components or isotopes) have been used. They can reveal spatio-temporal patterns of mixing in
the riparian zone and provide a quantitative estimate of mixing ratios (Appelo and Postma, 2005; Battin, 1999; Pinay
et al., 1998; Schilling et al., 2017; Stigter et al., 1998), as long as it is possible to properly identify the system end-
members (McCallum et al., 2010). Geostatistical methods have also been used to identify and understand the
distribution of different water sources within riparian zones (Lessels et al., 2016). However, these approaches rely on
intense water sampling for identifying the extent, to which different water sources mix (Biehler et al., 2020; Lessels
et al., 2016; Schneider et al., 2011). Yet, such methods still have limitations in capturing the full spatio-temporal
dynamics of SW-GW exchange and mixing in stream corridors. Assessing the spatio-temporal evolution of mixing
zones and their implications for the biogeochemistry of stream corridors remains a significant challenge.

The use of hydrodynamic models can yield detailed insights into stream-riparian zone exchange dynamics (Broecker
et al., 2021; Hester et al., 2017; Liu and Chui, 2018). In particular 3D fully-integrated surface-subsurface models that
explicitly account for SW-GW exchange fluxes at high spatial and temporal resolution, such as HydroGeoSphere
(HGS, Aquanty Inc., 2015) or ParFlow (Kollet and Maxwell, 2006) are well suited for this purpose. Still, most
numerical models cannot quantify the extent of different water sources solely based on computed water fluxes and
resulting water flow paths (Gomez-Velez et al., 2017). Such quantification usually requires additional solute transport
routines, and in turn extra computational resources, which can be facilitated via particle tracking techniques (Frei et
al., 2012; Nogueira et al., 2021b). The Hydraulic Mixing-Cell (HMC) method (Partington et al., 2011) is one such
approach that allows a quantification of mixing and can be applied to any hydrological model that provides an explicit
fluid mass balance at sufficiently resolved spatial scale (e.g., at the scale of numerical model cells). The method was

100 originally developed to identify the contribution of different water sources - namely surface water (e.g., surface runoff) and groundwater - to the total streamflow hydrograph (Gutiérrez-Jurado et al., 2019; Liggett et al., 2015; Partington et al., 2012, 2013), but it has also been applied to track water from different sources in other contexts such as groundwater abstraction (Schilling et al., 2017), or the spatio-temporal variation of mixing fronts (Berezowski et al., 2019).

### 1.3 Purpose of this study

In this study, we aim to map the different water sources and assess the dynamics of their macroscopic mixing within a riparian zone in order to evaluate the potential for biogeochemical turnover. To do so we use a state of the art numerical model and mixing cell routine. Rather than explicitly simulating the reactions induced by SW-GW mixing
(Hester et al., 2014, 2019), our objectives are to:

1. quantify the different water sources within the riparian zone and their spatio-temporal evolution;
2. assess the relationship between flow dynamics and the degree of macroscopic mixing of these different waters; and
3. evaluate the formation and dynamics of mixing hot-spots within the riparian zone.

To reach our objectives we do not aim to produce a meticulously calibrated, complex model for the studied site, but rather to harness the insights that detailed field observations in conjunction with such numerical modelling of macroscopic mixing provide (i.e., an "hypothetical reality") (Mirus et al., 2011). We again emphasize that here "mixing" refers to the colocation of different source waters within a defined volume of aquifer (e.g., a numerical model cell). Mixing degrees were computed based on transient results of HMC, which does not require further solute
transport simulations in order to track different water components in space and time. The HMC routine was coupled to a transient and fully-integrated 3D numerical flow model covering the riparian zone of a 4$^{th}$-order stream. We evaluate the HMC results in the light of hydrochemical data, and further quantify distinct mixing hot-spots that have the potential to enhance mixing-dependent turnover processes (Hester et al., 2014, 2019; Trauth et al., 2014). With this easy-to-transfer framework we also seek to demonstrate the utility of the HMC method for the identification of
mixing hot-spots at the river-corridor scale.

### 2 Methods

The steps followed in this study to assess the spatio-temporal variations of water sources and mixing within a riparian zone are summarized in Fig.2. In brief, following field data collection, a 3D numerical flow model was developed and
130 calibrated against the collected data (Nogueira et al., 2021b). The HMC method is then coupled to the numerical model, whereas results are additionally evaluated according to additional hydrochemical data (i.e., water samples) for the further mapping of water sources and analysis of mixing degrees in the riparian zone. In the subsequent sections we detail each step and methods followed.

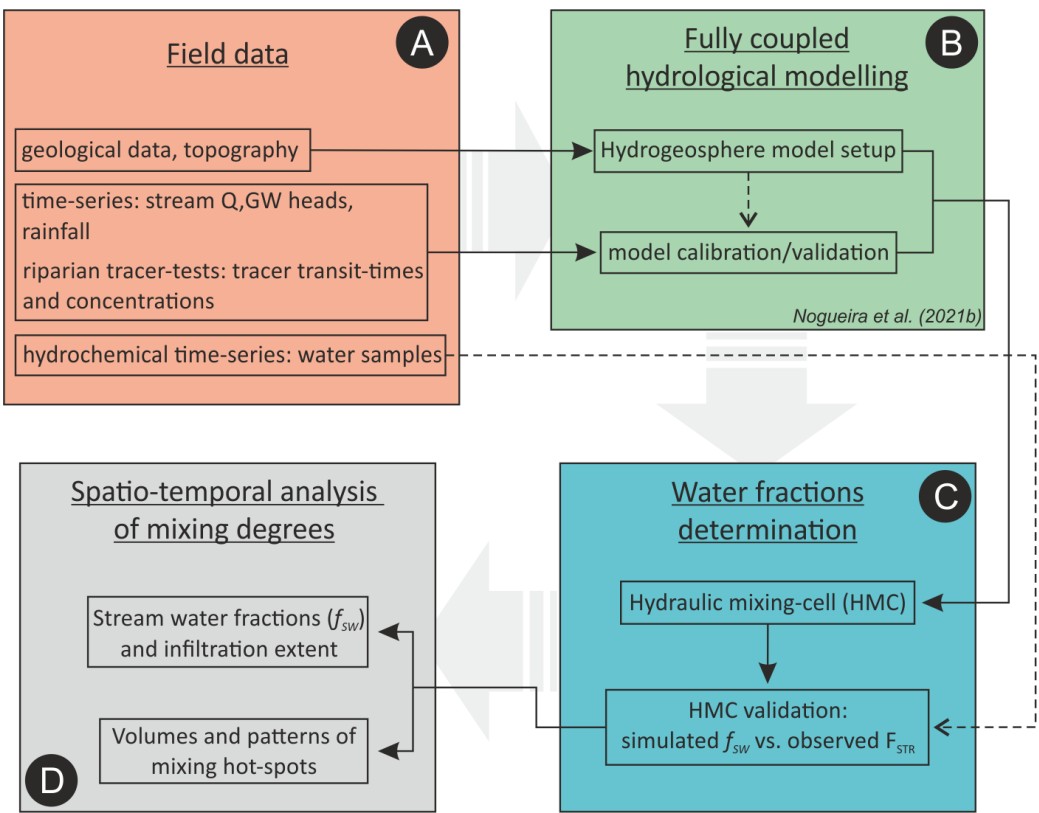

**Fig.2:** Flowchart of methods used to assess the spatio-temporal dynamics of the hyporheic zone and of the mixing degrees.

### 2.1 Study area and hydrological modelling

We coupled the HMC method to a previously calibrated numerical surface-subsurface flow model (Nogueira et al., 2021b) of a highly instrumented test-site of the TERENO observatory (Wollschläger et al., 2017). The study site is located within the catchment of the Selke Stream, a 4th-order perennial stream, in central Germany, Fig.3. The studied stream section (appx. 900 m) is characterized by predominantly losing conditions, which has been linked to enhanced turnover of groundwater-borne $NO_3^-$ at the site due to mixing with infiltrating stream-borne DOC and subsequent denitrification (Gassen et al., 2017; Lutz et al., 2020; Trauth et al., 2018). The alluvial aquifer consists of up to 8 m-thick fluvial sediments, with grain sizes ranging from medium sands to coarse gravels, underlain by less permeable clay-silt deposits forming its bottom. Other borelog and geophysical data reveals that the thickness of the alluvial aquifer steadily decreases with distance from the stream (Lutz et al., 2020; Trauth et al., 2018). The numerical flow model presented in Nogueira *et al.,* 2021b, which is based on the code HydroGeoSphere (HGS), is used here for coupling the HMC method since HGS explicitly computes fluid mass balances at every model cell and at each time-step of the simulation. HGS provides a fully-integrated 3D solution for variably saturated subsurface flow (using Richards's equation) and a 2D depth-averaged solution for surface flows based on the diffusive wave approximation to the St. Venant equations (Therrien et al., 2010).

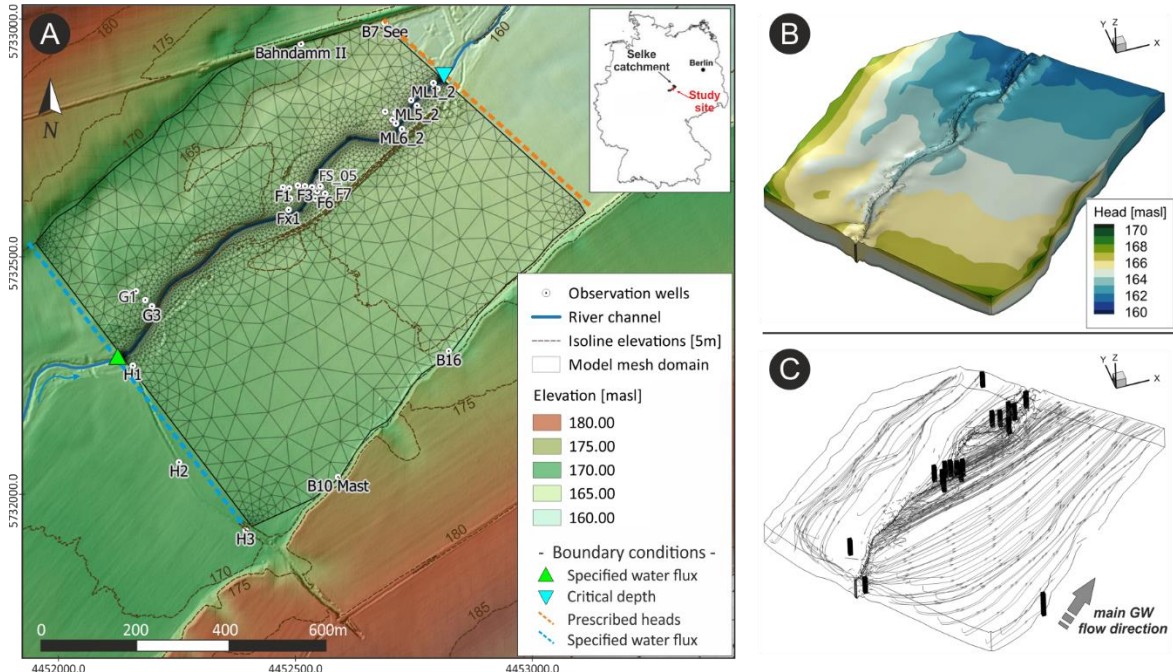

**Fig.3: a)** study area and model domain; **b)** simulated groundwater heads for a baseflow (Q=0.1 m³/s) scenario; **c)** streamlines (grey lines) depicting main groundwater flow direction for the baseflow scenario. Black vertical lines in (c) depict some of the wells shown in (a). Note the vertical exaggeration of the 3D plots (10x).

The flow model parameterization is only briefly summarized here, as the model and its calibration are described in detail in Nogueira *et al.* 2021b). The simulated domain ($900 \times 770 \times 10$ m) was divided into four main hydrogeological units according to geophysical and borelog data, which further indicates the thinning of the alluvial aquifer with distance from the stream (Lutz et al., 2020; Trauth et al., 2018; Zhang et al., 2021). Thus, the simulated domain covers most of the mapped alluvial aquifer present in the area. The bottom of the numerical model was set as a no-flow boundary in line with the less permeable clayey-silty deposits and the low-permeability bedrock at the base of the coarser alluvial sediments. The boundary conditions (BCs) on the model surface domain were defined as (i) groundwater recharge (as a fraction of daily precipitation) at the model top, (ii) specified water flux at the model stream inlet according to discharge values measured at a gauge station about 3000m upstream of the study site, and (iii) a critical depth BC at the model stream outlet (Fig.3a). The BCs on the subsurface model domain were defined as (iv) specified water flux representing ambient groundwater flow at the upstream side of the model, and (v) prescribed time-varying hydraulic heads at the downstream side of the model (Fig.3a). The other lateral subsurface boundaries of the model domain were set as no-flow boundaries based on field observations indicating that GW flowlines are somewhat parallel to the stream at this distance. The model was calibrated using the PEST software (Doherty, 2018) based on stream discharge values, multi-well groundwater heads, as well as multiple breakthrough curves from performed groundwater tracer-tests (Nogueira et al., 2021a, 2021b). Automatically calibrated parameters were within

the literature ranges and the calibrated model showed a very good match between observed and simulated values, with

coefficient of determination (R²) and Kling-Gupta-Efficiency (KGE) (Gupta et al., 2009; Knoben et al., 2019) generally above 0.8 and 0.5, respectively. The flow model was previously calibrated for the period of 2017-2018 and here we have only implemented changes in the BCs, without any additional model calibration. We performed transient simulations using daily forcing inputs for the hydrological years 2013-2016 since this is the period with more hydrochemical data available to further validate the HMC results.  The quality of the flow model was evaluated

according to the water balance error, R² between observed and simulated groundwater heads and stream discharge for the period under analysis (2013-2016), as well as KGE.

## 2.2 The Hydraulic Mixing-Cell (HMC) method

The contribution of water sources (and subsequent mixing degrees) in each model cell was calculated with the HMC

method (Partington et al., 2011, 2012, 2013). The different water sources have to be predefined in terms of their origin (e.g., stream water, groundwater, and rainfall), which are related to the BCs applied to the numerical model. HMC calculation depends only on computed nodal water fluxes and does not involve any extra parameters. The HMC method uses the ''modified mixing rule'', which simulates a mixing regime between perfect mixing and piston flow. Initially, all model cells have an artificial "initial" water fraction. In the subsequent time-steps, different water sources

are mixed according to volumes of water flowing into and out of a cell accordingly (Partington et al., 2011):

$$f_{i(w)}^t = \left( \frac{V_i^{t-1}}{V_i^t} - \frac{Vbc_{out}^t + \sum_{j=1}^m V_{ij}|_{t-1}^t}{V_i^t} \right) f_{j(w)}^{t-1} + \frac{Vbc_w^t + \sum_{j=1}^n V_{ji}|_{t-1}^t \, f_{j(w)}^{t-1}}{V_i^t} \tag{1}$$

where $f_{i(w)}^t$ [-] is the computed water fraction $w$ at time-step $t$ in cell $i$, $n$ and $m$ are sources and sinks for cell $i$, $f_{j(w)}^{t-1}$ denotes the water fraction $w$ at time $t$-$1$ in a neighbouring cell $j$, $V$ denotes the volume with the superscript denoting time-step and subscript $i$ denoting the cell, $ij$ denoting volume into cell $j$ from cell $i$ over the time-step from $t$-$1$ to t, $ji$

denoting volume from neighbour cell $j$ into cell $i$, and $Vbc_w^t$ is a volume from the inflowing boundary condition associated with water fraction $w$ and $Vbc_{out}^t$ is a volume summed from all outflowing boundary conditions at cell $i$. Inflow from adjacent cells is assigned the computed water fractions from the upstream cell. The HMC has an independent sub-time-step routine to calculate water fractions between the adaptive HGS time-steps, which circumvents the need of extremely small time-steps in the HGS simulations (Partington et al., 2013). This sub-routine

is required to avoid instability during the HMC calculations, which can occur if the volume of water leaving a cell over a time-step is greater than the volume in storage.

Within our simulations, we defined three main water sources to be tracked, namely stream water ($f_{SW}$), groundwater ($f_{GW}$), and floodplain water ($f_{FW}$). The $f_{SW}$ represents any water parcel that infiltrates into the subsurface domain through streambed cells; the $f_{GW}$ represents groundwater flowing into the domain through the upstream subsurface boundary;

the $f_{FW}$ represents water that percolates from soil top through the unsaturated zone (e.g., from rain or flood events). An additional water source named initial groundwater ($f_{GWi}$) was defined representing the "initial" water residing in

the model cells at the beginning of the simulations. We ran the model for a spin-up period at the beginning of the simulations in order to establish a more realistic distribution of the three water fractions over the domain at the beginning of our analyses. The spin-up period consisted of a two-year simulation period using constant average BC

values. Following this period, the fGWi fraction was virtually zero, whereas the three remaining water fractions were the only fractions observed throughout the domain. Thus, in the remaining analyses we mainly consider the three remaining water fractions for our calculations.

The sum of all HMC fractions in each model cell is $[f_{SW}+f_{GW}+f_{FW}]=1$, for an error-free fluid mass balance. With that approach, we can evaluate the composition of the different water fractions at any time-step and location at the model

domain. We further spatially aggregated the different HMC fractions to assess the temporal variation of their contribution to the total volume of the simulated domain with the *Integration* function in TecPlot 360 EX, Version 2019 R1 (TecPlot, Inc.) using the different HMC fractions as scalar variables and dividing the results by the total volume of the simulated domain. The function integrates the numerical cells within the simulated domain taking into account only the fraction of interest that comprises each cell volume. The calculation sums the resulting quantities

over the domain to produce the integrated result, which is then normalized by the total volume of the simulated domain ($V_{tot}$). Thus, the resulting volume $V_w$ represents a percentage of the total simulated domain:

$$V_w = \frac{\sum_{p=1}^{P}(V_p\, f_{w,p})}{V_{tot}} \times 100\% \tag{2}$$

where $V_w$ is the percentage volume of a HMC fraction *w* within the model domain in a given time-step, V is the volume of a model cell, *p* is a cell (from *p*=1 to P) with a specific water fraction $f_w$ (e.g., $f_{SW}$, $f_{GW}$, $f_{FW}$), and $V_{tot}$ is the

total volume of the simulated domain ($4.63\times10^6$ m³).

A similar version of Eq.2 was used to assess the spatio-temporal evolution of the hyporheic zone (HZ). To do so, we employ the *geochemical definition* of the HZ, similar to the one proposed by Triska et al. (1989), where the HZ is the area within the riparian zone containing more than 50% of stream water ($f_{SW} \geq 0.5$) in the mixture of waters. Using Eq.2, we computed the total volume of the HZ ($V_{HZ}$) in each time-step by only aggregating cells presenting $f_{SW} \geq 0.5$

in the domain. This *geochemical definition* was preferred over the *hydrodynamic definition* (Gooseff, 2010; Trauth et al., 2013) because of its stronger relevance for biogeochemical transformations (Boano et al., 2010; Gomez-Velez et al., 2017). Besides, in strongly losing streams, the HZ definition based on hyporheic streamlines (i.e., *hydrodynamic definition*) would describe the HZ as a very narrow zone limited to the streambed and its immediate vicinity only, while most of the infiltrating SW does not immediately return to the stream.


### 2.3   HMC validation and stream water fraction calculation

In order to validate the HMC results, we compared the simulated stream water fractions ($f_{SW}$) with the calculated stream water fractions ($F_{STR}$) at riparian observation wells. The $F_{STR}$ is based on a two end-member chloride (Cl⁻) linear mixing model (Appelo and Postma, 2005). By assuming Cl⁻ as a conservative solute, mixing between two

independent end-members occurs, namely stream water and groundwater farther away from the stream (not affected by infiltrating stream water). The fraction of stream water in the riparian groundwater was computed as:

$$F_{STR} = \frac{[Cl_{obs}^-] - [Cl_{GW}^-]}{[Cl_{SW}^-] - [Cl_{GW}^-]} \tag{3}$$

where $[Cl_{obs}^-]$ , $[Cl_{GW}^-]$, and $[Cl_{SW}^-]$ indicates the Cl- concentration measured in an observation well, in the groundwater distant from the stream, and in the stream at a given time, respectively. Calculations and measurements are based on biweekly collected water samples of 2014-2016. Groundwater was sampled with a peristaltic pump placed at the middle of the fully screened wells and surface water was collected as grab samples. Samples were stored and analysed in the lab following standard procedures (Trauth et al., 2018). The groundwater end-member $[Cl_{GW}^-]$ was assumed to be equal to values from the observation well B10 ($95 \pm 5$ mg L$^{-1}$, Fig.S1, supplementary material). To compare $F_{STR}$ and $f_{SW}$, we extracted simulated $f_{SW}$ values from the locations of the observation wells in the numerical model by averaging the $f_{SW}$ values of all fully-saturated cells that comprises each well position. That was done to approximate how water samples were collected at the fully screened wells, which likely results in sampling of a mix of the whole saturated column rather than from a specific groundwater depth. In a perfect model $F_{STR}=f_{SW}$ independent of the other simulated HMC fractions. The quality of the results was evaluated for each well in terms of the coefficient of determination ($R^2$) and with the nonparametric Wilcoxon rank-sum test (Ziegel et al., 2011) between $F_{STR}$ and $f_{SW}$ datasets. With the test, a result of *h=0* (null hypothesis) indicates the distributions of both populations are statistically equal. A value of *h=1* (alternative hypothesis) indicates the distributions of both populations are not equal.

### 2.4 Calculation and analyses of mixing degrees

### 2.4.1 Mixing degree calculation

At this stage, the results enable us to track and assess the different water source compositions at different time-steps and locations of our domain. We further computed a mixing degree (*d*) to quantify the degree to which different water sources mix within a model cell similarly to Berezowski *et al.*, (2019). We emphasize that the quantification of mixing here does not refer to true pore-scale mixing, but it rather gives an indication of how different water sources are "mixed" within a model cell in a given time-step based on neighbouring cells inflows and outflows. In that sense it provides a proxy for the potential for true pore-scale mixing to occur with that model cell.

For a three end-member mixing, where each end-member is a different water source (e.g., $f_{SW}$, $f_{GW}$, and $f_{FW}$), any three fractions combined could be represented by a vector in a 3D coordinate space: d=[$f_{SW}$, $f_{GW}$, $f_{FW}$], whereas a "perfect mixing" (e.g., equal fractions of different water sources) is represented by a vector $d_p$=[$^1/_3$, $^1/_3$, $^1/_3$] (Fig.S2, supplementary material). Thus, the resulting mixing degree *d* can be calculated as the Euclidean distance between the vectors *d* and $d_p$ taking into account that a maximum value for a given fraction can only be 1, as well as that the fractions have to sum up to 1 within a cell (for an error-free fluid mass balance). A more general equation to quantify the mixing degree for three (or more) end-members (w) could be written as:

$$d = 1 - \left[ \frac{\sqrt{\left(1/w - f_1\right)^2 + \left(1/w - f_2\right)^2 + \dots + \left(1/w - f_w\right)^2}}{\left(\sqrt{2} \times \sqrt{w}/w\right)} \right] \quad (4)$$

where $f_1$, $f_2$, and $f_w$ represent HMC fractions. Based on preliminary results, we have observed that actual volumes of $f_{FW}$ were very low in comparison to $f_{GW}$ and $f_{SW}$ in the fully saturated portion of the domain as it will be demonstrated in section 3.2. This occurs because recharge from rainfall is very low locally (Nogueira et al., 2021b), and the percolation of water from the top of the model domain is further limited to only occasional episodes. Therefore, we have employed a simplified version of the Eq.4 considering a two end-member mixing only. To do so, we combined the two end-members $f_{GW}$ and $f_{FW}$ to a single one (e.g., $[f_{GW}+ f_{FW}]$, Fig.S2, supplementary material), which reduces the mixing model to a two 2D case. This simpler two end-member mixing is the preferred one used throughout the manuscript because otherwise resulting $d$ values would consistently be very low in the simulations, which would impair a robust further analysis of the mixing fractions. In this formulation, $d=1$ represents a perfect mixing within a cell at a given time-step (e.g., equal water fractions: $f_{SW}=0.5$ and $f_{GW}+f_{FW}=0.5$), while smaller values would indicate a disproportional contribution of one or another water sources to the mixture (e.g., too much of one water source and too few of another). By calculating $d$ in every location and time-step, we can identify the model cells where the water sources of interest are mixed at equal proportions and assess its dynamics without depending on solute transport simulations.

To analyse the temporal variation of different mixing degrees, we spatially aggregated model cells presenting different $d$ values (e.g., $d > 0$, $d \geq 0.25$, $d \geq 0.50$, and $d \geq 0.75$) in each time-step and compared them to the total volume of the simulated domain (Eq.5), as well as to the total HZ volume (Eq.6) to assess their relative percentage in each time-step:

$$V_d = \frac{\sum_{p=1}^{P} (V_p \, d)}{V_{tot}} \times 100\% \quad (5)$$

$$V_{d\_HZ} = \frac{\sum_{p=1}^{P} (V_p \, d)}{V_{HZ}} \times 100\% \quad (6)$$

where $V_d$ and $V_{d\_HZ}$ are the percentage volumes of cells presenting a certain $d$ value (e.g., $d > 0$, $d \geq 0.25$, $d \geq 0.50$, and $d \geq 0.75$) within the model domain and within the HZ, respectively, in a given time-step, V is the volume of a model cell, $p$ is a cell (from $p=1$ to P) presenting a certain $d$ value, $V_{HZ}$ is the HZ volume according to Eq.2. Here, mixing hot-spots ($d_h$) are characterized by model cells presenting $d \geq 0.75$, as equally used in Berezowski *et al.* (2019) for delineating the active perirheic zone after the definition of Mertes (1997). We also assessed the temporal development of mixing hot-spots at the domain by comparing the peaks of $d_h$ values from Eq.5 with the peak of discharge events observed in the simulation period. So we can evaluate when mixing hot-spots occur in relation to flow dynamics and their magnitude of occurrence. We computed the Spearman's rank correlation to rank the metrics of discharge events (e.g., peak prominence, event duration, time-to-peak) that control the increasing in $d_h$.

### 2.4.2 Transit-times within mixing hot-spots

The development of mixing hot-spots is a good indication of the locations and moments where and when mixing-dependent reactions such as the turnover of groundwater-borne $NO_3^-$ due to pore-scale mixing with infiltrating SW can occur. However, since time is also a relevant variable for biogeochemical processes, it is equally important to know for how long a certain water parcel resides within mixing hot-spots. To quantify this time span, we defined exposure-time ($d_{h-\tau}$), as the time that a water parcel resides within defined mixing hot-spots along its transit through the riparian aquifer. We computed transit-times ($\tau$) based on a transient particle tracking analyses according to HGS flow model results (Nogueira et al., 2021b). Flow paths were extracted from each HGS time-step based on massless particles released from streambed cells and from the top of the model domain. A total of around 1,300 particles were released in each HGS time-step, capturing main groundwater flow directions and infiltrating SW flow paths.

For this analysis, we differentiated the flow paths in two categories: flow paths of SW that infiltrates and subsequently exfiltrates back to the stream within the simulated domain (called hyporheic flow paths), and water flow paths that do not exfiltrate to the stream within the simulated domain (called floodplain flow paths). Each flow path is divided into smaller sub-sections, which were analysed in terms of HMC fractions, $d_h$, $\tau$, and thus $d_{h-\tau}$. By carrying out these combined analyses, we can assess how $d_{h-\tau}$ is affected by transient hydrological conditions. Model visualization, integration and particle tracking analyses were performed in TecPlot 360 EX; additional calculations were carried out with MatLab® 2019b.

## 3 Results

In this section we will focus on the results of simulations for the years of 2013-2016 since it was the period used for the validation of HMC results. The results of the flow model are not detailed here, but only generalized for a better understanding of the SW-GW exchanges dynamics and overall characteristics of the flow system. Simulated groundwater heads and stream discharge matched the field values well, with a mean KGE of 0.73 for groundwater heads and 0.84 for stream discharge (Fig.S3, supplementary material). The stream reach was characterized by predominantly losing conditions with average net water losses to the subsurface of around 40-50% of total discharge. This is higher than the 25% measured in the field by Schmadel *et al.* (2016) during a small discharge event in July 2014; however, our simulated net water losses for the period of their analyses were around 30%, indicating a good match to observed reach conditions during the discharge event.

A small gaining portion was observed for the simulated reach only at a localized deep pool downstream in the domain (representing only about 1% of the total infiltrating SW), whereas the majority of infiltrating SW exited the domain via the downstream subsurface boundary. During very high discharge and overbank flow (generally $Q \geq 7.0$ m³s$^{-1}$) the near-stream riparian zone can be partially flooded. Moreover, groundwater flow paths are somewhat more parallel to the stream under low discharge and more divergent under high discharge conditions (Nogueira et al., 2021b).

### 3.1 Validation of HMC fractions

Before further assessment of the HMC results, the simulated stream water fractions ($f_{SW}$) were compared to observed stream water fractions ($F_{STR}$), which were calculated based on Cl⁻ measurements for a validation of model results (Sect. 2.3). The $F_{STR}$ computed according to Eq.2, as well as the extracted $f_{SW}$ for some observation wells are presented in Fig.4. The locations of the wells are presented in Fig.3.

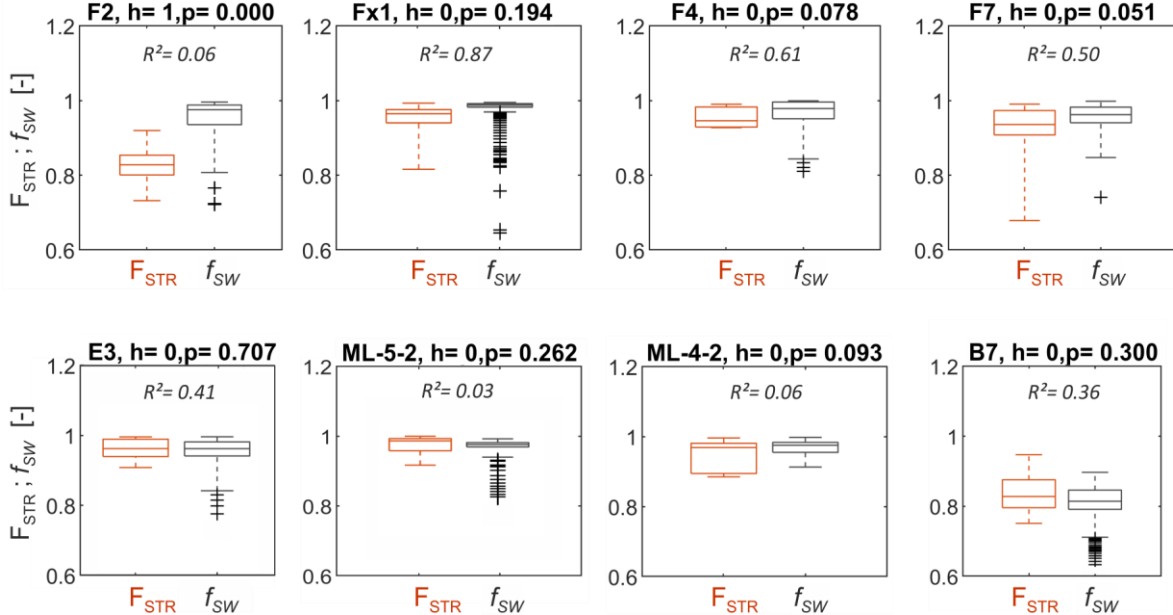

**Fig.4:** Observed and simulated stream water fractions ($F_{STR}$ and $f_{SW}$, respectively) for some observation wells in the study area. The *h* values represent the results of the Wilcoxon-test between $F_{STR}$ and $f_{SW}$ datasets with respective *p-values* (p): h=0 indicates that the $F_{STR}$ and $f_{SW}$ groups are from continuous distributions with equal medians, while h=1 indicates the difference between the medians is statistically significant; The R values show the coefficient of determination between the $F_{STR}$ and $f_{SW}$ datasets. The names of the wells are shown at the top of each plot.

For 70% of all groundwater samples the mixing model was applicable for the calculation of $F_{STR}$. For the other samples, Cl⁻ concentrations were temporally lower than in the stream water and they were excluded from further analyses (Fig.S1, supplementary material). In general, the observation wells had exhibited high $F_{STR}$ values, indicating higher fractions of stream water than other components like groundwater, Fig.4. Calculated $F_{STR}$ and simulated $f_{SW}$ for wells presented similar ranges (between 0.7 and 1.0), while $F_{STR}$ showed slightly larger variations in comparison to $f_{SW}$ values. Despite that, correlating calculated $F_{STR}$ with simulated $f_{SW}$ showed reasonable coefficients of determination ($R^2$ values shown in Fig.4) indicating that the model generally captures the variations of stream water fractions in the riparian groundwater for most observation wells. Small differences between $F_{STR}$ and $f_{SW}$ existing in some of the wells (e.g., ML wells and well F2) can be related to localized processes and conditions not captured by the model as later discussed in the manuscript. The Wilcoxon-test performed between $F_{STR}$ and $f_{SW}$ datasets individually for each

observation well also indicated that the populations were not statistically different for the majority (indicated by h=0 on Fig.4), reinforcing the good match between simulated and observed stream water fractions on riparian groundwater.

### 3.2 Spatio-temporal variation of simulated HMC fractions

The temporal variation of simulated HMC water fractions (here referred to as just "fraction(s)") is presented in Fig.5.
A spin-up period required to flush the initial $f_{GWi}$ was found to be around 2 years, which is an approximation to the time required to fill the aquifer with "new" water sources. From this point on throughout the manuscript, we focus only on the analyses of the three remaining fractions (i.e., $f_{SW}$, $f_{GW}$ and $f_{FW}$).

Integrating the different fractions over time (Eq.2), on average 35% of the simulated domain comprised water originating from the stream ($f_{SW}$), whereas groundwater inflowing from the upstream subsurface boundary ($f_{GW}$) was
around 35%, and 30% consisted of water originating from the soil surface ($f_{FW}$). Since the HMC results indicate the water-origin rather than the water content, we further evaluate the HMC results considering only the fully-saturated portion of the model domain using Eq.2, which we can then relate to total HMC water contents in the subsurface. In terms of stream water, nearly 80% of the saturated domain presented $f_{SW} \geq 0.1$, and about 20% presented $f_{SW} \geq 0.9$. Following the geochemical HZ definition ($f_{SW} \geq 0.5$), this corresponded to around 50% of the fully-saturated domain
(Fig.5b). Likewise, 50% of the fully-saturated domain consisted of surface water, followed by 40% consisting of groundwater and only 10% of floodplain water, Fig.5c. This indicates relatively small contributions of water originating from the top of the domain to the saturated portion of the aquifer for most of the simulated period.

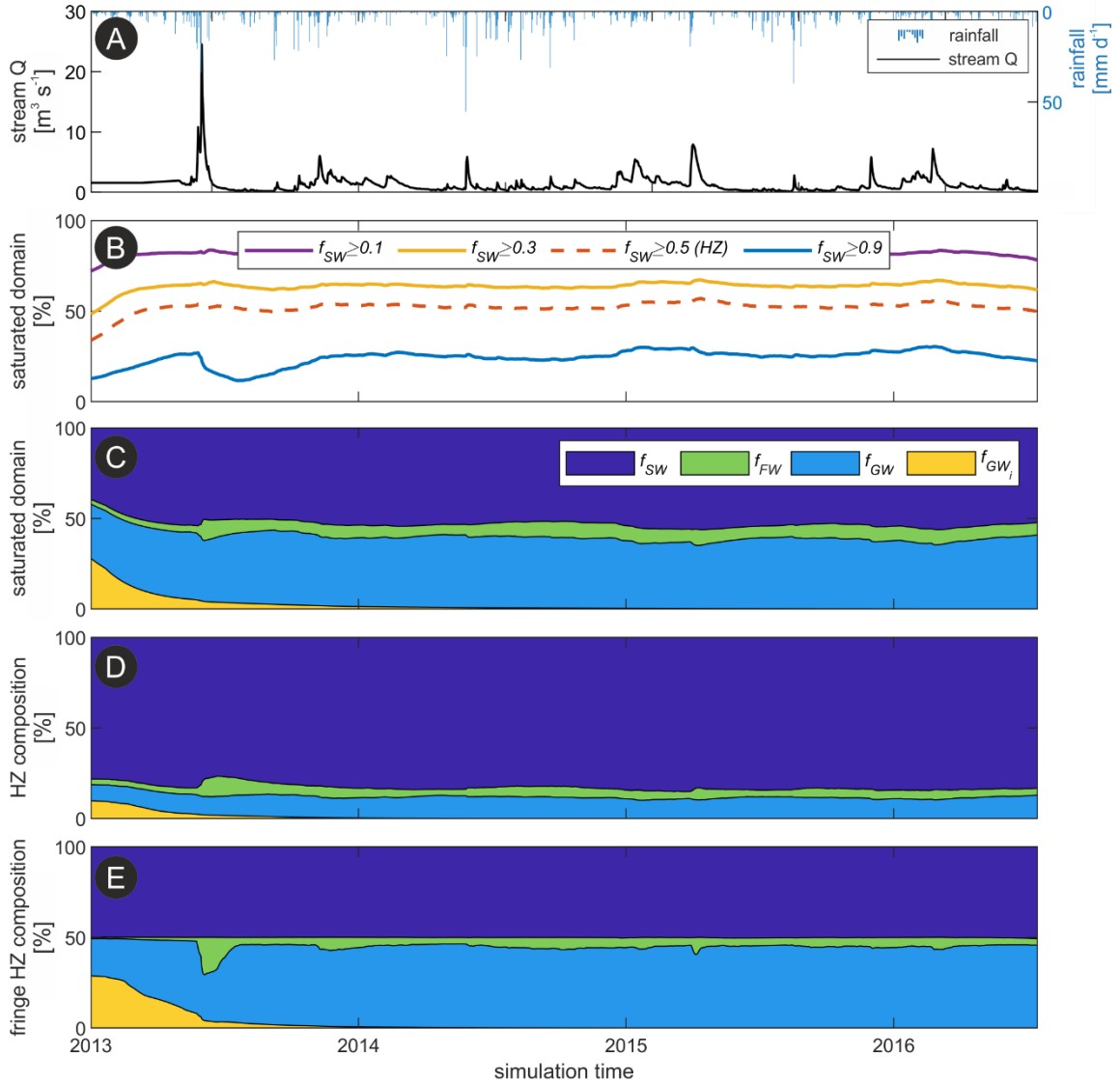

**Fig.5: a)** Stream discharge (Q) and rainfall time-series; **b)** temporal variation of the saturated domain consisting of at least a certain fraction (e.g., 0.1, 0.3, 0.5, and 0.9) of stream water ($f_{SW}$); **c)** contribution of different fractions to the saturated domain (stream water ($f_{SW}$), floodplain water ($f_{FW}$), groundwater ($f_{GW}$), and initial groundwater ($f_{GWi}$)); **d)** composition of different fractions to the hyporheic zone (HZ, $f_{SW} \geq 0.5$); **e)** composition of different fractions at the fringe of the HZ ($f_{SW}=0.5$). Note that the start of the simulation (when $f_{GWi}=100\%$ and the other water fractions are zero) is not shown in the plot.

Around 80% of the geochemical HZ volume consisted of water originating from the stream, with the rest being represented by groundwater (15%) and floodplain water (5%), Fig.5d. This already suggested that, despite the potential for subsurface biogeochemical processes and turnover of stream-borne solutes within hyporheic flow paths (Trauth et al., 2014; Zarnetske et al., 2011), there is limited potential for mixing-dependent reactions involving reactants from both water sources (SW and GW) due to the dominance of stream water in this zone. Differently, at the HZ fringe (where $f_{SW}=0.5$), $f_{GW}$ and $f_{FW}$  40% and 10% respectively (Fig.5e), indicating a higher potential for mixing between the different water sources.

The Fig.6 shows the spatial distribution of minimum, maximum and median values, as well as the standard deviation (σ) of each fraction within the domain for the entire simulation period. The plots indicate the minimum and maximum possible distributions of each water fraction in the domain, as well as their typical distribution throughout the simulation period. To better represent the maximum probable HMC water contents (and not only the proportions of the different sources) in each model cell, the fractions shown in the plot were multiplied by the maximum saturation value that was recorded in each model cell during the entire simulation period.

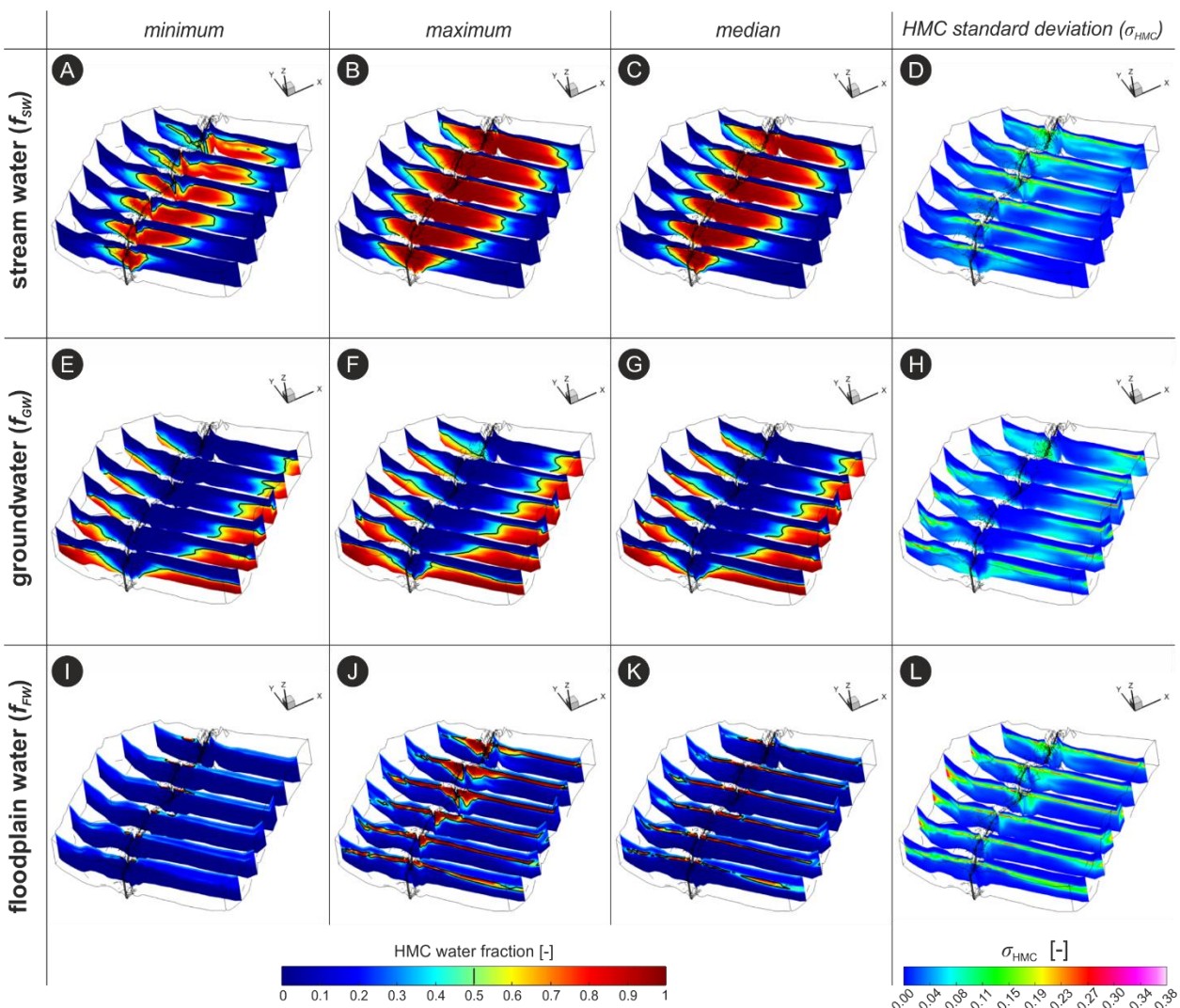

**Fig.6:** slices throughout the simulated 3D domain showing the minimum, maximum, median values, as well as standard deviations ($\sigma_{HMC}$) of stream water ($f_{SW}$) **(a-d)**, groundwater ($f_{GW}$) **(e-h)**, and floodplain water ($f_{FW}$) **(i-l)** fractions for the entire simulation period in different segments of the domain. The black line **(a-c, e-g, i-k)** indicates the HMC fractions of 0.5. Note the vertical exaggeration of the 3D plots (20x).

Throughout the simulation, $f_{SW}$ was high around the stream and decreased with distance from the stream, Fig.6a-c, reaching values of 0.5 at around 150-200m from the stream channel, which defines the local geochemical HZ. This regularity was maintained by the continuous SW infiltration to the aquifer due to the overall losing conditions of the stream reach. The $f_{SW}$ plume was slightly smaller at upstream areas due to boundary condition effects. There was also a large variation of $f_{SW}$ values around the stream and at the groundwater-table interface (Fig.6d). High values of $f_{GW}$ were only observed at the periphery of the simulated domain, as well as at the southern upstream face of the domain (boundary effect), Fig.6e-g. Lastly, high $f_{FW}$ values were mainly observed above the groundwater-table. However, since absolute water content (i.e., saturation) is low in this portion of the domain, the total $f_{FW}$ content is also relatively low in comparison to other HMC fractions, Fig.6i-k. Still, some high $f_{FW}$ values were recorded in the subsurface (Fig.6j), when a considerable volume of water originating from the stream flows overbank and subsequently percolates through the riparian soils (e.g., following the high discharge event on Jun-2013, Fig.S4, supplementary material). Although the model indicates this is floodplain water, it is important to keep in mind that it is overbank flow of stream water that subsequently percolates into the subsurface after flooding.

### 3.3 Spatial variation of mixing degrees

Despite the nearly constant spatial distribution of fractions throughout the domain (Fig.6), flow dynamics and stream stage fluctuations resulted in different mixing degrees between the different fractions. The plots in Fig.7 show the spatial distribution of minimum, maximum and the median values of mixing degree, as well as its standard deviations ($\sigma_d$) for the entire simulation period (2013-2016). The plots show the minimum and maximum possible distributions of the mixing degrees in the domain, as well as their average distribution for the entire simulation period.

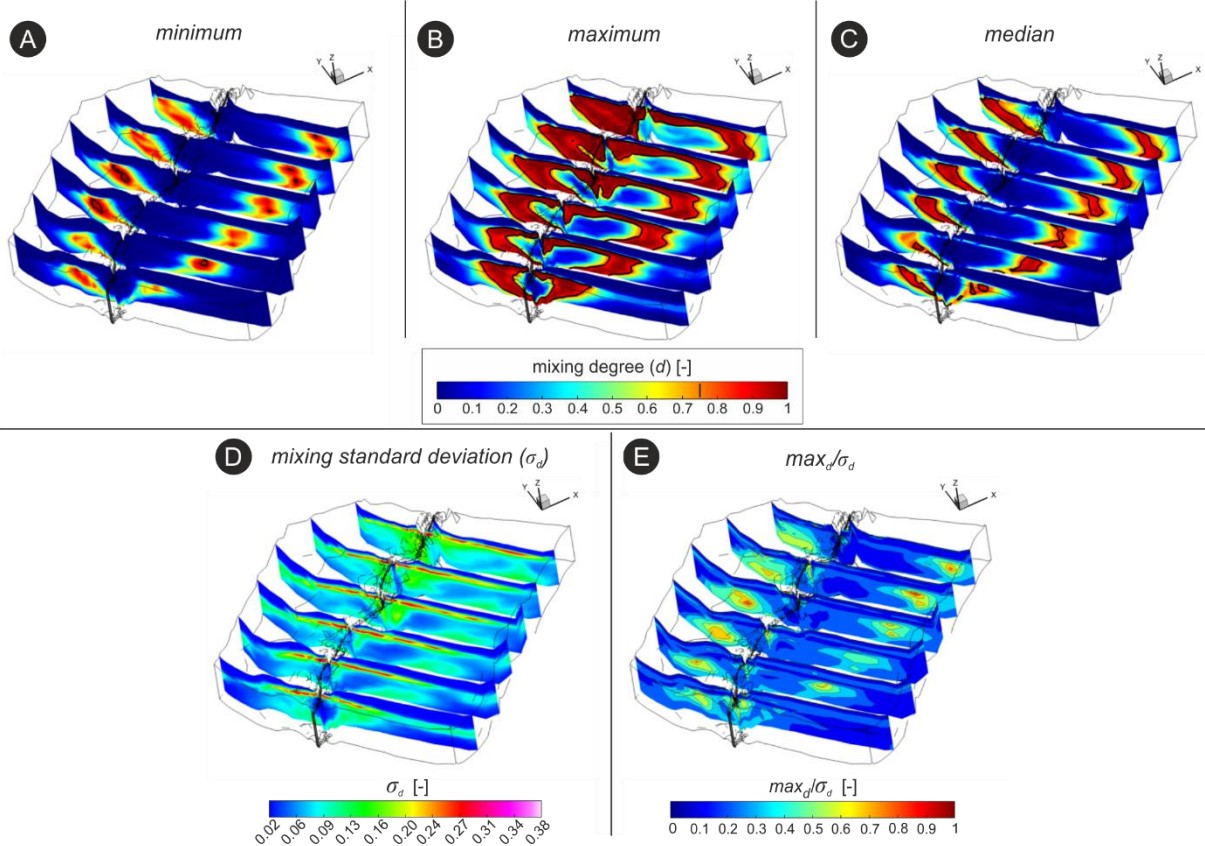

**Fig.7:** slices throughout the simulated 3D domain showing the minimum **(a)**, maximum **(b)**, median **(c)**, standard deviation ($\sigma_d$) **(d)**, and the normalized ratio $max_d/\sigma_d$ **(e)** of mixing degrees for the entire simulation period in different segments of the domain. The black lines **(a-c)** indicate regions with *d=0.75* (*mixing hot-spots*, $d_h$). Note the vertical exaggeration of the 3D plots (20x).

The relatively high $f_{SW}$ within the HZ prevents high mixing degrees to occur near the stream. In contrast, regions at the fringe of the HZ presented the highest minimum *d* values over the entire simulation period (Fig.7a), which suggest constant high *d* values in those areas. Yet, larger *d* values also occurred near the stream at some points during the

simulation (Fig.7b). These large *d* values near the stream followed discharge events with partial flooding of the riparian zone, which leads to large percolation of inundation water into the riparian aquifer, which then mixes with infiltrating SW and with ambient groundwater. Nevertheless, the computed median values of *d* indicate that *mixing hot-spots* ($d_h$) ($d \geq 0.75$) were indeed more persistent near the HZ fringe (Fig.7c) throughout the simulation. In comparison to regions near the groundwater-table interface, for example, these areas at the HZ fringe also presented slightly smaller $\sigma_d$

(Fig.7d), showing smaller variation in time. We further quantified the persistence of *mixing hot-spots* in time by computing and normalizing the ratio of maximum *d* over their $\sigma_d$ ($max_d/\sigma_d$) since a small $\sigma_d$ alone does not imply a persistent *mixing hot-spot* over time. A high value of this metric would indicate the occurrence and persistence of *mixing hot-spots* over the entire simulation period, as it can be observed near the HZ fringe for instance (Fig.7e) where $max_d/\sigma_d$ are generally above 0.5. These areas comprise only around 5% of the total model domain.

### 3.4 Temporal variation of mixing degrees

Mixing degrees also varied in time, as could be concluded from the plots in Fig.7. Here, we further assessed how mixing degrees varied over time, as well as their relationship with flow dynamics. For that, we have integrated cells with at least a certain degree of mixing (e.g., $d > 0$, $d \geq 0.25$, $d \geq 0.50$, and $d \geq 0.75$) and compared them to the total volume of the domain (Eq.5). Around 80% of the domain presented some sort of mixing ($d > 0$), which strongly varied over time suggesting the activation of areas that do not present consistent mixing throughout the simulation, Fig.8b. Zones with $d \geq 0.25$ were on average 40% of the total domain. Only around 9% of the total domain presented $d \geq 0.50$, which was just slightly larger than $d_h$ ($d \geq 0.75$). Moreover, $d_h$ represented 7-12% of the total domain volume (Fig.8c). In relation to the geochemical HZ, *mixing hot-spots* were comparatively higher and represented on average 23% (between 15% and 30%) of the total HZ volume (Fig.8d).

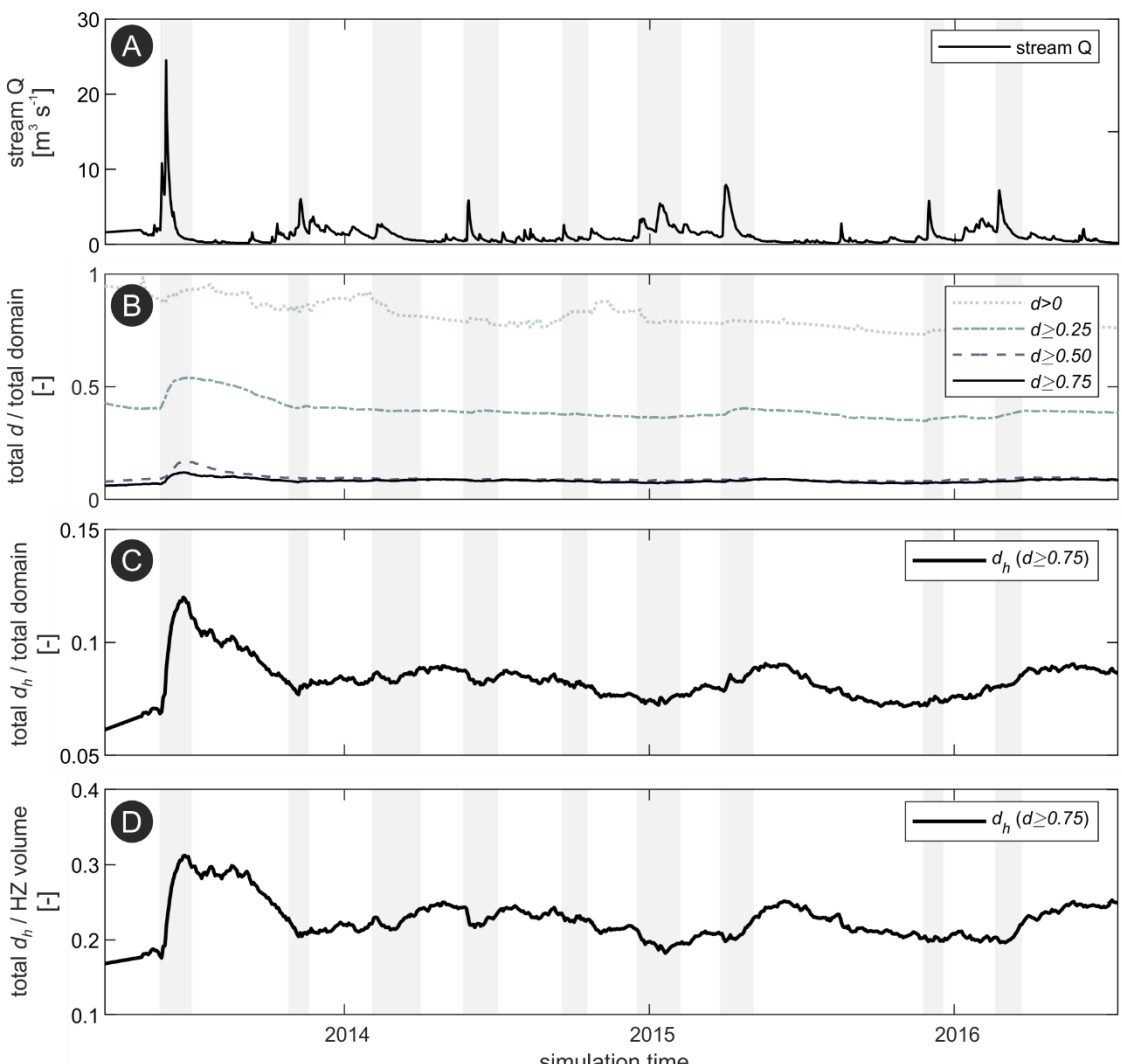

**Fig.8: a)** time-series of stream discharge for the period of 2013-2016; **b)** total volume of cells presenting a certain degree of mixing ($d>0$, $d≥0.25$, $d≥0.50$, and $d≥0.75$) in relation to total domain volume; **c)** total volume of mixing hot-spots ($d_h$, $d≥0.75$) in

 relation to total domain volume; and **d)** total volume of $d_h$ in relation to total hyporheic zone (HZ) volume. Grey vertical bars indicate discharge events periods.

The impacts of discharge (Q) variations on $d$ were also evident (Fig.8b). Concerning *mixing hot-spots*, discharge events increased $d_h$ by 5-10% in comparison to conditions immediately before the start of the events (Fig.8c). Spearman's rank correlation (Table 1) showed that the discharge peak prominence ($\Delta Q$) (in relation to Q value prior to the event) and the increase of $d_h$ from the value immediately prior the event ($\Delta d_h$) were positively correlated ($R_{spear}=0.96$). Both the event duration and the time-to-peak were not strongly correlated to $\Delta d_h$ ($R_{spear}=0.09$ and $R_{spear}=0.30$, respectively) (Fig.S5, supplementary material). In our simulations, event duration and peak prominence were also not strongly correlated ($R_{spear}=0.14$, data not shown).

Moreover, the lag between the peak of the discharge events and the peak of $d_h$ was 14 days on average, somewhat shorter for events presenting higher $\Delta Q$, but the metrics showed only a weak correlation ($R_{spear}=0.28$). On the other hand, event duration and the lag between the peak of the discharge events and the peak of $d_h$ showed a good correlation ($R_{spear}=0.66$), suggesting that longer events would lead to later developments of $d_h$. Due to the temporal lag between the peak of Q events and peak of $d_h$, mixing degrees were generally higher during the recession of discharge events.

**Table 1:** Overall Spearman's rank correlation between metrics of discharge events and the increasing of mixing hot-spots ($d_h$) at the riparian zone.

| Discharge events metrics | Correlation to $\Delta d_h$ |
|---|---|
| Event duration [days] | 0.009 |
| Time-to-peak [days] | 0.305 |
| Time-to-peak/event duration [-] | 0.340 |
| Peak prominence ($\Delta Q$) [m³ s⁻¹] | 0.963 |

| Lag between Q *peak-event* and following peak $d_h$ [days] | |
|---|---|
| Min | 1 |
| Mean | 14 |
| Max | 46 |

$R_{spear}$ between $\Delta Q$ and lag to peak $d_h$: 0.28
$R_{spear}$ between event duration and lag to peak $d_h$: 0.66

### 3.5  Exposure-times ($d_{h-\tau}$)

Since the time that a water parcel resides within *mixing hot-spots* also affects the potential for biogeochemical processes, for each flow path we computed exposure-time ($d_{h-\tau}$), as the share of water transit-times ($\tau$) spent within *mixing hot-spots*. Overall, $d_{h-\tau}$ were generally smaller during the peak of discharge events since: i) groundwater velocities are higher during events, leading to relatively shorter $\tau$ (Fig.9a), and ii) $d_h$ was relatively smaller during peak events (Fig.8c). Since transit-times are generally longer under baseflow conditions (Fig.S6, supplementary material), $d_{h-\tau}$ was equally longer during the recession of discharge events (Fig.9b).

Specifically, the median $d_{h-\tau}$ of floodplain flow paths (i.e., water parcels that do not exfiltrate through streambed cells within the model domain) was highly variable in time (3-12 days), on average 15% of the total flow path $\tau$, Fig.9b. On the other hand, for the hyporheic flow paths (i.e., infiltrating SW that exfiltrates through streambed cells after infiltration and subsurface transit), $d_{h-\tau}$ were small (0-3 days), on average 5% of the total hyporheic $\tau$. The median $\tau$ of floodplain flow paths were slightly stronger correlated to Q variations than hyporheic flow paths, $R_{spear}$= -0.50 and $R_{spear}$= -0.45, respectively (Fig.9c).

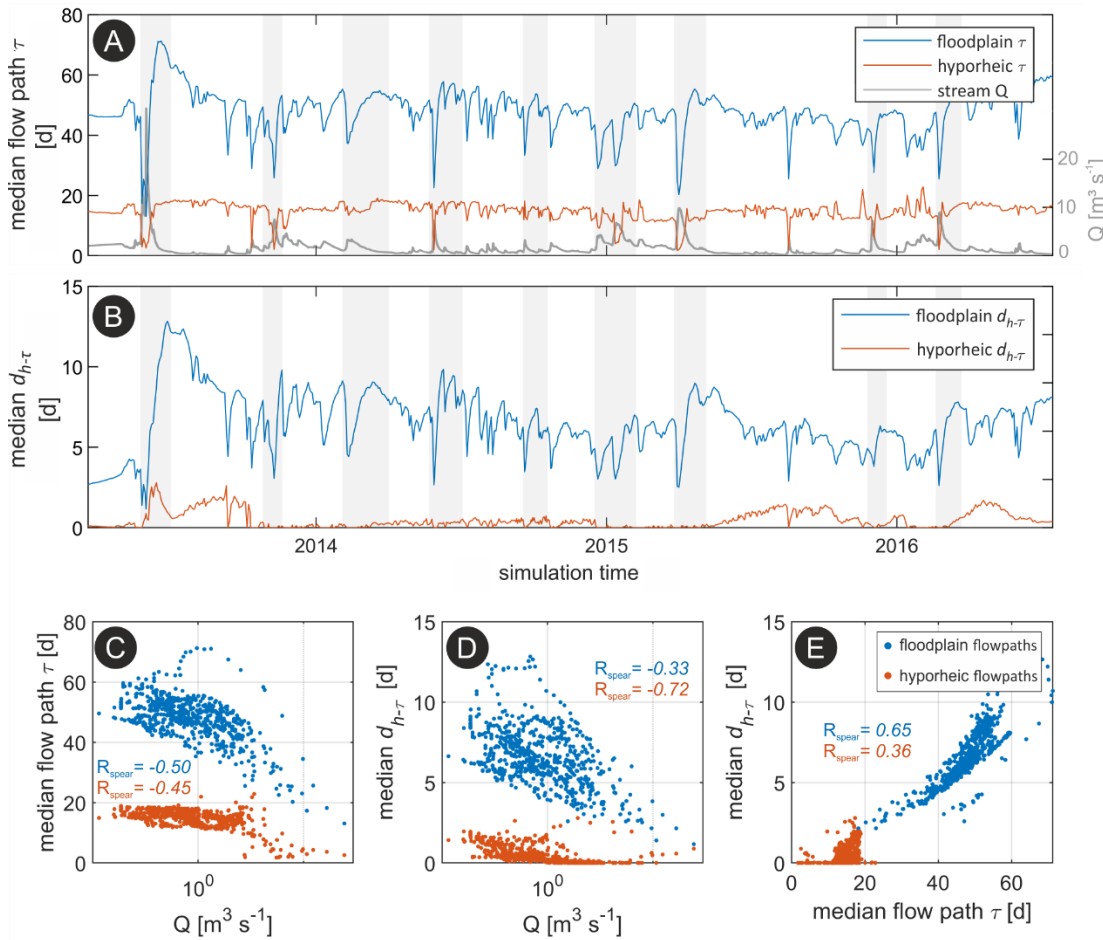

**Fig.9: a)** median transit-times (median flow path $\tau$) alongside stream Q; **b)** median exposure-times (median $d_{h-\tau}$); **c)** median flow path *vs.* stream discharge (Q); **d)** median $d_{h-\tau}$ *vs.* Q; and **e)** median $d_{h-\tau}$ *vs.* median flow path $\tau$. The Spearman's rank correlation ($R_{spear}$) between variables is showed in the scatter plots (c-e). Grey vertical bars (a-b) indicate discharge events periods. Note the log scale for Q values in (c-d).

The Fig.9b indicates that hyporheic $d_{h-\tau}$ increases under baseflow conditions relative to values during discharge events although hyporheic $\tau$ were somewhat constant over time (Fig.9a). Indeed, hyporheic $d_{h-\tau}$ were inversely correlated to stream discharge ($R_{spear}$= -0.72, Fig.9d), but only weakly correlated to variations of transit-times ($R_{spear}$= 0.36, Fig.9e). In contrast, for floodplain flow paths, $d_{h-\tau}$ was only slightly negative correlated with stream Q ($R_{spear}$= -0.33), whereas

they showed a stronger correlation with flow path transit-times ($R_{spear}= 0.65$). Whereas both hyporheic and floodplain $d_{h\text{-}\tau}$ decrease with increasing Q due to overall shorter water transit-times, the occurrence and controls of mixing hot-spots due to flow dynamics at these different regions are somewhat different, as it will be discussed in Sect. 4.3.

## 4 Discussion

### 4.1 Validation of the flow model and the HMC results

In this study, we coupled a previously calibrated numerical flow model with the HMC method (Partington et al., 2011) in order to assess the distribution of different water fractions in a stream corridor, using the riparian zone of the Selke stream as a study case. The numerical model used here had been calibrated based on another observation period (Nogueira et al., 2021b), but after implementation of correct hydrological BCs (e.g., stream inflow, groundwater heads at the boundary) showed good agreement with field data from the period investigated in this paper. This reinforces the quality of the original calibration and justifies the application of the numerical flow model to another time period after BC adjustments. Small mismatches between observations and simulated values in terms of groundwater heads and stream discharge could be related to the simplified geology within the numerical flow model, which can affect SW-GW dynamics and groundwater flow paths (Fleckenstein et al., 2006; Gianni et al., 2019; Savoy et al., 2017), as well as to the simplified streambed heterogeneity that can modify overall SW-GW exchange fluxes (Pryshlak et al., 2015; Tang et al., 2017).

Usually, numerical flow models are solely calibrated based on hydrological observations. Previous studies using the HMC method have rarely attempted to validate their results based on hydrochemical data, exceptions being the studies by Liggett *et al.* (2015) and Berezowski *et al.* (2019), for instance, while this could further enhance model reliability and parameterization (Partington et al., 2020; Schilling et al., 2017, 2019). Here, in addition to groundwater heads and stream discharge evaluation, we verified the HMC results by comparing simulated $f_{SW}$ and calculated $F_{STR}$, the latter based on a Cl$^-$ mixing model. The calculation of $F_{STR}$ was possible for most of the water samples (70%), whereas a few of them presented unrealistic $F_{STR}>1$ due to Cl$^-$ concentrations being temporally lower than in the stream water end-member. We attribute this to local variability in evaporation or the presence of geogenic Cl$^-$ that can affect Cl$^-$ concentrations (Delsman et al., 2013; Ong et al., 1995). Nevertheless, the simulated $f_{SW}$ values matched the field $F_{STR}$ values at the observation wells quite well, indicating a good performance of the HMC method for mapping water source composition despite model simplifications. The small differences between $F_{STR}$ and $f_{SW}$ were acceptable given that a calibration to hydrochemical data was not performed in this study, and that the model captured the main SW-GW dynamics and hydrochemical variations, well which is further discussed in Sect. 4.5. These results suggest that the HMC method can be a valuable tool, complementary to more labour-intensive field sampling, for mapping patterns of water source composition and their temporal variation at the riparian zone and watershed scales (Berezowski et al., 2019; Schilling et al., 2017).

### 4.2 HMC fractions and HZ dynamics

In terms of water origin there was a nearly constant distribution of the different HMC fractions within the model domain (35% stream water, 35% groundwater, and 30% floodplain water). However, taking a look at the fully-saturated domain only (70-80% of the total simulated domain), reveals that 90% of the water in the saturated zone originates from the stream (50%) and from groundwater flowing into the domain via the upstream boundary (40%), Fig.5a. This is manifested in a geochemical HZ (region presenting $f_{SW} \geq 0.5$) that extends up to 200 m into the riparian aquifer. While this may appear as a large percentage of stream water in the riparian aquifer, other studies of alluvial aquifers reported equally large percentages of stream water in the riparian aquifer at large distances from the stream (up to 250m), which were especially controlled by the permeability of the aquifer material (Schilling et al., 2017). Similarly, Poole *et al.* (2008) have found that alluvial aquifer water at the Minthorn study site (gravel-alluvial dominated aquifer) was essentially all derived from the main stream channel of the Umatilla River. They also found that the *geochemical* HZ penetrates to the entire local riparian zone (about 300m wide) (Jones et al., 2008). In contrast, Sawyer *et al.* (2009) estimated the HZ extent to be only up to 30 m from the banks of the Colorado River near the Hornsby Bend site. In their case, however, the hydraulic conductivity of the aquifer material was nearly one order of magnitude smaller than on the other sites and on ours (and the stream reach has not been predominantly losing). Those previous studies and our findings go in line with the propositions of Boulton *et al.* (1998) and Wondzell (2011) on the combined influence of hydrogeology and stream flow dynamics on the development of the hyporheic zone and exchanges around streams. Both studies have suggested that the cross-sectional area of the hyporheic zone relative to the stream channel are the highest for low-order streams, while unconstrained lowland streams present the greatest hyporheic zone cross-sectional area relative to wetted stream channel (Boulton et al., 1998).

Within the local HZ, most of the water was advected stream water (approximately 80%). This is also in line with previous studies highlighting the dominance of purely advected surface water within hyporheic zones (Hester et al., 2014, 2019). Within the simulated domain, most of the infiltrated stream water did not immediately return to the stream and may therefore be termed "groundwater" after some transit through the aquifer. However, the fact that it originated from the stream manifests in a different chemical composition compared to ambient groundwater. For instance, the infiltrated stream water will have higher contents of dissolved oxygen (DO) and dissolved organic carbon (DOC) compared to ambient groundwater (Trauth et al., 2018). In turn, the mixing between the infiltrated SW and ambient GW, can deliver DOC as an electron donor to facilitate denitrification of groundwater-borne nitrate (Hester et al., 2019; Song et al., 2018; Trauth et al., 2014; Trauth and Fleckenstein, 2017).

### 4.3 Variations and controls of mixing degrees and mixing hot-spots

Only a few studies attempted to quantify the spatio-temporal variations of the mixing degrees resulting from SW-GW exchange process at the stream-corridor scale (Lessels et al., 2016), as well as their potential implications for biogeochemical processes. While some studies have relied on extensive field campaigns (Gassen et al., 2017; Jones et al., 2014), numerical simulations carried out by Trauth & Fleckenstein (2017) and Hester et al. (2019), suggested the importance of mixing zones for the denitrification of groundwater-borne nitrate. Here, on average, nearly 50% of

the model domain presented $d \geq 0.25$ throughout the simulation period. About 9% of the domain (and roughly 20% of the HZ) could be defined as *mixing hot-spots* ($d \geq 0.75$), with most of them being located at the fringe of the HZ. The persistence of these mixing hot-spots in time could be illustrated with the metric $\max_d/\sigma_d$, which was consistently high at the fringe of the HZ. This is qualitatively consistent with previous smaller-scale studies showing that mixing hot-spots between SW-GW tend to occur in narrow zones at the fringe of the HZ (Hester et al., 2013; Sawyer and

Cardenas, 2009; Trauth and Fleckenstein, 2017). Likewise, Berezowski *et al.* (2019) computed $d_h$ values slightly above 6% of the total area of a larger basin in Poland following a large flood event, $d_h$ that were also mainly located at the fringe of the HZ.

In our simulations, magnitudes of peak discharges during events were strongly correlated with increases of $d_h$ over the event. This is in line with Trauth & Fleckenstein (2017), who found that for the same event duration, discharge

events with higher peaks increased denitrification of groundwater-borne nitrate by a factor of up to 7x due to enhanced mixing with stream-borne DOC. In the same way, Hester et al. (2019) showed that the size of the SW-GW mixing zone below a streambed dune increased and shifted with increasing SW depth (analogous to increasing stream discharge in this study). While our results indicated a similar expansion of the mixing zones following discharge events (Fig.8c), we could also observe and quantify the temporal shift of $d_h$ peaks (e.g., counter-clockwise hysteresis

with a peak of stream discharge events) alongside the shift of their locations within the riparian zone.

Water transit-times are usually used as a metric to assess the HZ reactive potential since the longer the transit-time, the higher the potential for solute transformations (Boano et al., 2010; Zarnetske et al., 2011). To evaluate this potential in relation to reactive mixing zones we defined the exposure-time ($d_{h-\tau}$), as the time water resides within model cells classified as mixing hot-spots. Our results show that the hyporheic $d_{h-\tau}$ were generally smaller than (non-hyporheic)

floodplain $d_{h-\tau}$ and more negatively correlated with stream discharge (Fig.9d). This is mainly because hyporheic transit-times are generally shorter than floodplain water transit-times. Besides, under low stream discharge conditions, ambient groundwater flow is somewhat more parallel to the stream (Nogueira et al., 2021b), while groundwater flow towards the stream increases due to a decrease in SW depth (Buffington and Tonina, 2009). This and the slightly stronger gaining conditions at the pool located further downstream in the model domain (Fig.S4, supplementary

material) result in a greater SW-GW mixing near the stream region, hence increasing the hyporheic $d_{h-\tau}$. With increasing stream discharge, however, SW influx into the riparian aquifer increases, which shifts the SW-GW mixing front to regions farther from the stream (Hester et al., 2019) and hyporheic $d_{h-\tau}$ decreases.

In contrast, with distance from the stream, $d_{h-\tau}$ is mainly controlled by variations in water transit-times. This is because mixing far from the stream is mainly enhanced by increasing stream discharge, which brings SW to farther distances

within the aquifer where it can mix with ambient groundwater. In line with our results, Trauth et al. (2015) found the total consumption of groundwater-borne nitrate within an instream gravel bar to be higher under neutral and slightly gaining ambient groundwater conditions (i.e., low stream discharge). This is when the total influx of solutes from the stream is low, but consumption of groundwater-borne nitrate is high due to enhanced SW-GW mixing. Previous work on hyporheic reactivity has often been carried using 1D or 2D model setups focusing on biogeochemical processes in

direct vicinity of the streambed (Hester et al., 2014, 2019; Newcomer et al., 2018). This study, using a larger-scale 3D model also considers lateral SW-GW exchange fluxes over longer distances into the riparian aquifer the associated longer-term mixing processes further away from the stream channel. In line with results from Nogueira et al. (2021b) and Trauth et al. (2018), results from our 3D model coupled with the HMC method reinforce that such larger-scale and long-term processes are important around losing streams for the creation of mixing hot-spots at larger distance

from the stream. These mixing hot spots can facilitate mixing-dependent biogeochemical reactions, which may significantly contribute to the net turnover of groundwater-borne solutes at the stream corridor scale. These processes may have been overseen in small-scale studies, which have focused on the immediate interface between the stream channel and the alluvial groundwater only.

Likewise, our results suggest that discharge events can enhance turnover of groundwater-borne solutes in the riparian

zone at locations farther from the stream more than in the hyporheic regions near the stream. Conversely, under low discharge conditions, hyporheic $d_{h\text{-}\tau}$ increase due to slightly increasing GW upwelling and subsequent SW-GW mixing. Nevertheless, in strongly gaining stream reaches with a dominance of GW-seepage to the stream (e.g., limited or absent hyporheic flow paths), hyporheic transit-times (Cardenas, 2009; Trauth et al., 2013, 2014), as well as SW-GW mixing (e.g., in terms of flux magnitude) (Hester et al., 2013; Sawyer et al., 2009) would be smaller, and

consequently the potential for turnover of groundwater-borne solutes would be smaller too (Hester et al., 2019).

### 4.4    Mixing hot-spots and biogeochemical implications

In order to further show the implications of mixing for local biogeochemical processes, we compared our HMC results with hydrochemical analyses from Gassen et al. (2017), who monitored water quality across the groundwater-table interface using a local multilevel piezometer that can be sampled at highly resolved depth intervals in the variably

saturated vadose and fully saturated groundwater zones (Fig.S7, supplementary material).

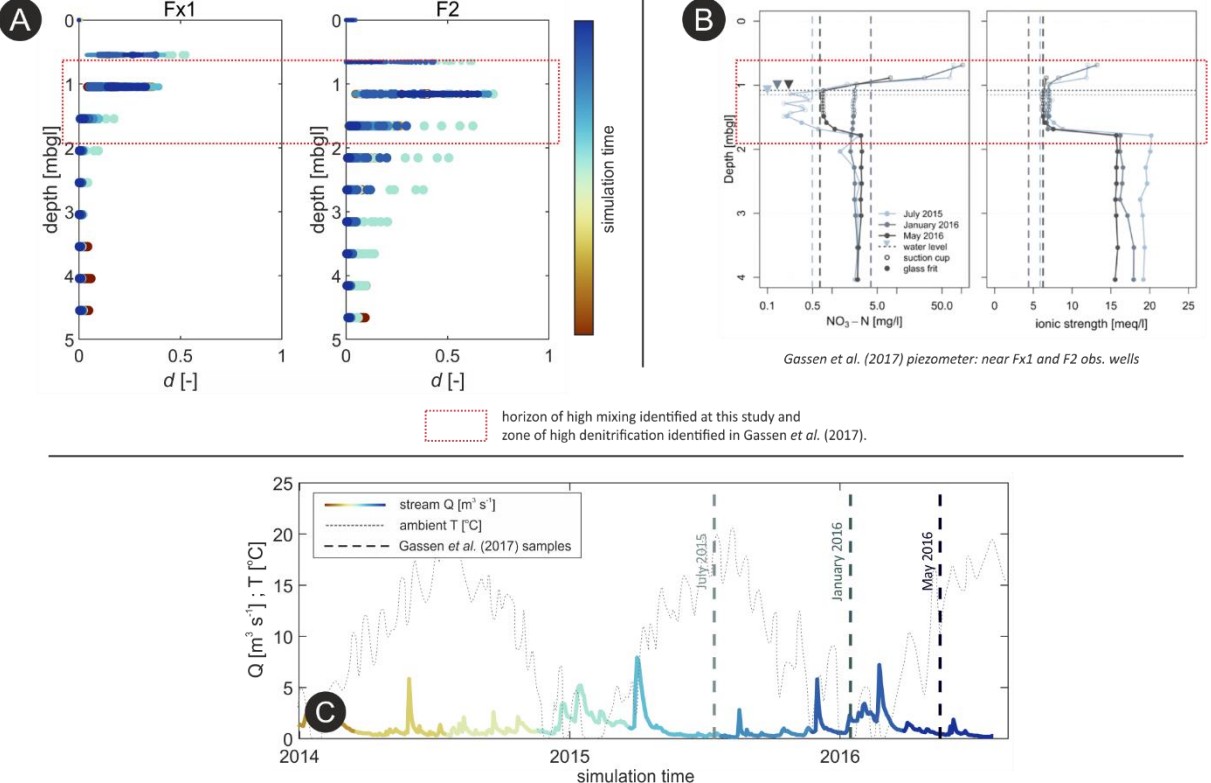

**Fig.10: a)** mixing degrees with depth for two observation wells. Colours indicate simulation time; **b)** measurements of NO3−N concentrations and ionic strength in a multilevel piezometer for three different sampling dates. Vertical dashed lines represent NO3−N concentrations/ionic strength of the stream water, horizontal dotted lines represent the groundwater-table at the corresponding sampling date (reprinted (adapted) with permission from Gassen et al. (2017), Copyright (2017), American Chemical Society); and **c)** ambient temperature alongside stream Q coloured according to simulation time. Vertical dashed lines indicate the sampling dates from Gassen et al. (2017).

The HMC-simulated vertical variations of mixing degrees at the near stream observation wells show strong similarity with observed vertical variations in nitrate concentrations and a denitrification fringe around the water table separating high concentrations in the vadose zone form significantly lower concentrations in the saturated zone as highlighted by the red rectangle in Fig.10a and Fig.10b. Our simulations revealed generally higher mixing degrees ($d \geq 0.5$) over the top 1-2m of the saturated zone, while Gassen et al. (2017) observed high nitrate concentrations above the groundwater table (up to 70 mg L$^{-1}$), which exponentially decreased across the uppermost saturated zone to values below 3 mg L$^{-1}$. Besides seasonal temperature effects on denitrification rates (Fig.10c) (Nogueira et al., 2021a; Widdowson et al., 1988; Zheng et al., 2016), mixing with stream-borne DOC and subsequent denitrification is most likely the processes responsible for the observed high denitrification rates at the vadose zone-groundwater interface in the uppermost parts of the saturated zone. This reinforces the importance of mixing hot-spots for biogeochemical processes in riparian zones and highlights the importance of mapping different water sources and their mixing dynamics.

**4.5   Limitations of the employed method and recommendations for future studies**

Even though the numerical model matched field observations well, it represents a simplification of reality (a characteristic inherent to all models), which in turn results in some limitations and uncertainties. For instance, based on available geophysical data we have assumed the clayey-silty formation on top of the vertically tilted low-permeability bedrock as the bottom of the alluvial aquifer and impermeable in the model. We assumed that the alluvial aquifer has a limited lateral extent (Lutz et al., 2020; Trauth et al., 2018), which was backed by geophysical data and the presence of bedrock outcrops along parts of the lateral model boundaries. These assumptions and the chosen model geometry, however, may not fully account for larger-scale hydrological fluxes, which are inherent to nested SW-GW systems. For instance, as showed by Flipo et al. (2014) and by other studies (Boulton et al., 1998; Magliozi et al., 2018; Toth 1963), SW-GW system are connected interfaces, which are linked to each other through different spatio-temporal processes. For instance, longer and deeper flowpaths that might have been not represented in our numerical model could lead to the development of additional mixing spots at greater depths or distances from the stream (Lessels et al., 2016). This could further emphasize and explain how alluvial aquifers and riparian zones act as buffer zones connecting low-frequency processes occurring at regional scale and high-frequency processes occurring in the stream network (Ebeling et al., 2021; Flipo et al., 2014; Rivett et al., 2008; Sun et al. 2017). Equally, lateral influx of groundwater through the lateral boundaries of the model domain could also effects the dynamics and main directions of GW flow paths and therefore SW-GW mixing spots development. However, head data at the site did not show any indications of such effects. Furthermore the specific geology of the site with relatively shallow, low-permeability Mesozoic bedrock strata, which inhibit lateral groundwater movement as they are vertically tilted, rules out the presence of a laterally extensive, continuous regional aquifer. Exchange fluxes between the shallow alluvial aquifer with deeper groundwater were therefore considered to be negligible.

Despite good agreement with field $F_{STR}$ values, simulated HMC water fractions such as $f_{SW}$ were not included in the calibration of the numerical model. In a more rigorous calibration, this could have been done, which might further minimize mismatches between simulated and observed HMC fractions, while still respecting the parameter range. It is a trade-off with computation time since model calibration can largely increase with sub-routines for the calculation of observations/parameters of interest. Since the numerical model used here was previously calibrated based on both conventional and more unconventional oservations, and since the goal of this study was not to reproduce all details at the field site, we did not carry out additional model calibration. However, the addition of unconventional observation-types to model calibration (on the top of commonly used groundwater heads and stream stage/discharge measurements) tends to lead to a more robust calibration reducing equifinality in the parameter sets (Nogueira *et al.,* 2021b; Schilling *et al.*, 2017, 2019; Partington *et al.*, 2020), and should be considered in future studies.

We intentionally did not conduct explicit simulations of reactive transport in this study since our main goal here was to explore the HMC method (coupled to a flow model) to assess the development of mixing spots in the riparian zone and their relation to hydrological variations. Spatial patterns of mixing hot-spots can provide a meaningful proxy for the interpretation of reactivity patterns in the absence of extensive data for the parameterization of an explicit reactive transport model. Along those lines we could illustrate the importance of such macroscopic mixing spots for groundwater-borne $NO_3^-$ turnover by comparing the quantitative mixing results of the HMC method with previous

biogeochemical assessments carried out in the study area. For a direct quantification of nitrate removal rates, however, the use of reactive-transport models or additional field data combined with data-drive analyses would be needed. Such simulations would have allowed a comparison of observed and simulated concentration values and their dynamics for a more rigorous evaluation of model performance (Nogueira et al., 2021b). However, the additional computational effort to numerically solve the transport equations would likely also increase computational costs. Our model results matched patterns of mixing degrees estimated from field observations very well and the simulated patterns allowed an improved interpretation of observed processes. Furthermore our results were well in line with other studies on biogeochemical processes related to SW-GW mixing at comparable sites. The identification of hot-spots for macroscopic mixing between SW-GW with the HMC method can provide a good proxy for the occurrence of potential biogeochemical hot-spots for mixing-dependent turnover of groundwater-borne solutes in river corridors. However, care should be taken in interpreting such results as this "potential" may not be realized if stream-borne reactants (like DOC) have been exhausted before reaching the mixing hot-spots.

Finally, the HMC method is based on water fluxes computed between model cells and therefore assumes that all HMC fractions are perfectly mixed within a model cell at every time-step (Partington et al., 2011). This condition may be violated, if stratification of different waters exist over the vertical extent of a model cell (Karan et al., 2013; Kolbe et al., 2019). Although the vertical extent of the model cells in our study is much smaller than the extent over which significant stratification would commonly be assumed to occur, high-resolution local observations (e.g., of vertical concentration variations) may not be captured with our approach, which integrates over the scale of larger model cells.

## 5   Conclusion

Riparian zones contain waters from different sources, which can mix with each other and in turn enable mixing-dependent biogeochemical processing. In this study, we coupled a hydraulic mixing cell (HMC) method with a previously-calibrated transient and fully-integrated 3D numerical flow model to assess the distribution of different water sources in a riparian aquifer, as well as their mixing dynamics. The simulated mixing degrees matched estimated values based on natural chloride tracer data well. A qualitative comparison of HMC based mixing patterns with concentration patterns from additional, hydrochemical data generally confirmed the robustness of the method, which is computationally comparably cheap, as it does not require explicit solute transport simulations to track different water sources in space and time.

Our estimations indicated that along the simulated stream reach, about 50% of the water in the riparian aquifer originates from the stream, whereas about 40% is groundwater and the remaining 10% is floodplain water (e.g., from rainfall or flooding from top soil). This overall composition was relatively steady over time, but it was episodically affected by larger stream discharge events, which deliver larger volumes of stream water to the riparian aquifer via infiltration or overbank flow. Similarly, macroscopic mixing, evaluated in terms of the mixing-degrees, was observed at least in 80% of the domain ($d > 0$), but it was spatially and temporally variable within the riparian zone. On average, about 9% of the model domain could be characterized as mixing hot-spots ($d \geq 0.75$), but this percentage could be nearly 1.5 times higher following large discharge events. Moreover, event intensity (event peak magnitude) was found

to be more important for the increase of the spatial extent of mixing hot-spots than event duration. Our modelling results also indicate that event-driven changes in the fluxes and velocity of infiltrating stream water, affect exposure-times (i.e., time of a water parcel residing within a *mixing hot-spot*) along hyporheic flow paths to a larger extent than the exposure-times of water flowing far from the stream. With distance from the stream, exposure-times become increasingly controlled by variations in general water transit-times. In contrast, in the near stream zone, the rapid increase of SW influx during events shifts the ratio between the water fractions to SW, reducing the extent of potential mixing zones inhibiting mixing dependent reactions. At the same time increasing stream water infiltration at higher flow velocities delivers stream water further into the riparian aquifer, shifting the zones with significant macroscopic mixing between SW and GW away from the near stream zone.

The analysis of water source dynamics, and of the relationship between the mixing of different water sources and flow dynamics in a riparian zone presented in this study provides an easy-to-transfer approach for the mapping of water sources and the identification of *mixing hot-spots* within riparian zones. Understanding the patterns and dynamics of macroscopic mixing between SW and GW in riparian zones can help to better understand patterns of reactive turnover or the redistribution of other, non-reactive solutes or small particulate substances (e.g., micro plastic particles) in the riparian zone. Future assessments could also focus on smaller scale streambed mixing processes, considering, for instance, (1) more heterogeneous hydraulic conductivity fields at the streambed and at the riparian aquifer, as well as (2) different events duration and peak magnitudes.

## 6 Data availability

The field data, and the numerical model files are publically available and can be assessed through: https://doi.org/10.4211/hs.a0dc51142fe249f89877c4005e0b6947.

## 7 Author contributions

GN performed the formal analysis, the investigation and wrote the original draft of the manuscript; all the authors contributed to review, final writing and editing; GN, CS, and JF conceptualized the study; GN, DP, PB, and JF conceived the methodology; GN, CS, DP, and JF worked on the validation of the study; CS and JF were responsible for supervision; JF was responsible for funding acquisition, resources and project administration.

## 8 Acknowledgment

This research was financially supported by the ENIGMA-ITN project within the European Union's Horizon 2020 research and innovation programme under the Marie Sklodowska-Curie Grant Agreement No.722028, as well as the Collaborative Research Centre 1357 MICROPLASTICS funded by the German Research Foundation (DFG) Project

Number 391977956 – SFB 1357 and the Research Program of the Helmholtz Association. Special thanks to Tomasz Berezowski for his support in the study, as well as to two anonymous reviewers for their comments that helped to increase the quality of the manuscript

755

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
