# Peer review of "Spatio-temporal variations of water sources and mixing spots in a riparian zone"

_Hydrology and Earth System Sciences, 2021_

## Author Comment (AC1)

Nogueira et al presented a modeling study for estimating the water sources and mixing spots in a riparian zone. The authors developed a SW-GW coupling model and applied it in a 4th-order stream, then compared against hydrochemical monitoring data. With the modeling quantification, the authors show that discharge events increase the mixing and present that their tool can serve as a complementary approach for hotspot identification. I feel this paper is novel and well-written so I encourage publications after some of my comments are resolved.

We thank the reviewer for the constructive comments on our work, which have helped to improve our manuscript. We also recognize that some points were not clear and may have been confusing. We have revised the manuscript to clarify those points based on the comments provided by both reviewers. Below we provide our detailed responses to the reviewer's comments.

L92: The authors mentioned other options of models such as ParFlow, and others. To my knowledge I think HydroGeoSphere is a commercial model and is not open-source - Why would you use it? I noticed that the code development paper is in press on WRR, and I'm wondering how much reproducibility of this work from a commercial model? For example, did the authors reuse codes or functions from HydroGeoSphere? Is this work of code development going to be publicly available?

The reviewer is correct, HGS is a commercial software, which initially started as a research code. Hence the routines used in the code are scientifically established and have been widely documented in the scientific literature (e.g., Therrien and Sudicky 1996, Jones et al. 2008, Unger et al. 2008, Park et al 2009, Brunner and Simmons 2011). We chose HGS as a code due our long-term experience with the code, which dates back to earlier, pre-commercial times of initial model development, the numerical efficiency of the code and its ease of use as well as the codes direct interface to the HMC model. For a detailed implementation of the HMC code in the HGS, we refer to Partington et al. (2011). Although HMC can be easily used in HGS the use of this method is not exclusive to HGS. It could be used together with any other numerical code that provides an explicit water balance at every cell in the model domain (Partington et al., 2011), such as MODFLOW for example. With our study we seek to demonstrate the applicability and usefulness of the HMC method in mapping water sources in a riparian zone, regardless of the software employed for the flow simulations. Our proposed methodology involving the HMC routines could be easily adapted to and employed in other non-commercial groundwater flow software.

L94: I think here should follow up with a few particle tracking paper introduction to say that particle tracking is the method without using the solute transport.

We appreciate this suggestion. We have added some references in the respective sentence to better exemplify this concept.

L247: Is this equation 4) defined by the authors or from other literatures? If the former, a more detailed explanation may be needed. If the latter, a citation will be needed.

We apologize for the confusion. The equation is based on the equation presented by Berezowski et al., 2019 (cited on L242). We have modified the sentence to clarify this. We have also added a figure to the supplementary material to facilitate a better understanding of the concept (see now Fig.S2 – also attached at the end of this document).

In brief, for a three end-member mixing case, any combination of three different source water fractions can be represented as a point $d$ in a 3D coordinate space, in which the maximum Euclidian distance between point d and the point of equal mixing (equal fractions of all mixing members) within the mixing space is the radius of a circle (centred at [1/3, 1/3, 1/3]) escribed on an equilateral triangle (side length of $\sqrt{2}$). For a two end-member mixing case, the maximum segment is the diameter of a circle (centred at [0.5, 0.5]) with side length of $\sqrt{2}/2$. For a four (or more) end-member mixing case a spatial representation is not possible but equation (4) would equally apply. Therefore we would like to keep it in the text.

L297: Is Schmadel et al (2016) a study at the same research region? Would the discharge event affect the calculation? I feel it is not a direct comparison between this modeling result with other studies, unless more justification is needed. Also, what is "average losses" and "simulated losses" being defined?

Yes, Schmadel et al. (2016) performed their study at the same reach that we have performed our simulations at. With "average losses" we refer to the net water losses computed from discharge measurements at two different points along the reach (e.g., upstream and downstream in the domain), as similarly carried out by Schmadel et al. (2016).

We believe discharge events could affect this calculation because the reach could present slightly different net losses during some events in comparison to baseflow conditions, for instance, due to different hydraulic gradients between SW-GW, which result in different SW-GW exchanges. By using the water balance resulting from our simulations, we can compare our simulated water losses with their calculated value. We apologize for the confusion; we have now rewritten the sentence and changed the terms to water losses in order to better explain what we are referring to. It now reads:

> *The stream reach was characterized by predominantly losing conditions with average net water losses to the subsurface of around 40-50% of total discharge. This is higher than the 25% measured in the field by Schmadel et al. (2016) during a small discharge event in July 2014; however, our simulated net water losses for the same period of their analyses were around 30%, indicating a good match to observed reach conditions during the discharge event.*

L350: I'm more or less confused about the GW initial component in this plot. If GW initial term indicate the water budget from initial condition, does the F_GW term should be 0?

The reviewer is correct. At the very beginning of the simulations, the initial GW fraction ($f_{GWi}$) is equal to 1 in all model cells, while all the other water fractions are equal to 0. Therefore, a "spin-up" period is carried-out to flush out this virtual initial water and have a more representative

distribution of the other water fractions of interest throughout the domain. In the plots, we do not present this initial, "pre-spin-up" condition since our analyses focused on the remaining three water fractions. We have rewritten the figure caption to try and make this clearer. We have also added a sentence to clarify this in the methods section. It now reads (end of section 2.2):

> *We ran the model for a spin-up period at the beginning of the simulations in order to establish a more realistic distribution of the three water fractions over the domain at the beginning of our analyses. The spin-up period consisted of a two-year simulation period using constant average BC values. Following this period, the $f_{GWi}$ fraction was virtually zero, whereas the three remaining water fractions were the only fractions observed throughout the domain. Thus, in the remaining analyses we mainly consider the three remaining water fractions for our calculations.*

Fig.5: Still, I'm trying to understand the fraction response to a rainfall event, for example, summer 2013. I see that fSW actually decreases when this new rainfall event occurs - could you explain why and is this something you are expecting?

Thank you for bringing up this point for discussion. In this case, we could first observe an increase of $f_{SW}$ at the peak of the discharge event resulting from the large volume of stream water pushed into the riparian aquifer due to the increase of SW depth in the stream channel. This is then followed by a general increase of $f_{FW}$ (during the falling limb of the event) as pointed out by the reviewer. This due to larger portions of stream water overflowing the banks during the discharge event, which then re-infiltrates into the subsurface during the recession of the flow event. This reinfiltrating water is considered to be part of the $f_{FW}$ fraction within the simulation. Therefore, even though discharge is great during this event, there is a relatively larger increase in the $f_{FW}$ in comparison to $f_{SW}$ following the event.

This is something we expect in case of large discharge events due to flooding of the near stream area (Fig.S4, for example). For smaller discharge events, in which overbank flow does not occur, we expect an increase of $f_{SW}$ only, as it can be observed in other periods of the simulation.

Fig.6: I think the figure caption does not explain why multiple profiles are shown in each plot? This is confusing to me - what each slice represents?

We apologize for the confusion. In the plots, the slices are cuts throughout the 3D domain (instead of showing an opaque volume or a fully translucent object), which facilitates the visualization of the subsurface properties in our opinion. In each subplot, the slices represent the same properties, however in different segments of the subsurface. We have rewritten the captions of the figure (and of Fig.7 as well) to better explain this idea. For instance Fig.6 now reads:

> *Fig.6: slices throughout the simulated 3D domain showing the minimum, maximum, median values, as well as standard deviations ($\sigma_{HMC}$) of stream water ($f_{SW}$) (**a-d**), groundwater ($f_{GW}$) (**e-h**), and floodplain water ($f_{FW}$) (**i-l**) fractions for the entire simulation period in different segments of the domain. The black line (**a-c, e-g, i-k**) indicates the HMC fractions of 0.5. Note the vertical exaggeration of the 3D plots (20x).*

L526: Why is Spearman's rank correlation used instead of Pearson's correlation, which I think is more widely used?

We prefer to use the Spearman's rank correlation instead of the more commonly used Pearson's correlation, because with the later we primarily want to characterize and quantify the monotonic relationships between the parameters independent of the magnitudes of the changes in parameter values. As we do not expect linear relationships between the analysed parameters we considered the Spearman's rank correlation to be the more straight forward and robust metric to characterize the monotonic relationships. However, we attest that by using the Pearson's correlation coefficient, one would likely obtain somewhat similar values and ranges in the final correlation coefficients.

L635: I know a solute transport model is not yet developed so it is difficult to estimate how much computational saving by using this method. This can be an open questions I guess, but how do you justify the computational saving?

Thank you for bringing up this point for discussion. Indeed, particularly for the modelling area, this could be seen as an open question at the moment. However, an explicit solute transport model at the same site was used in Nogueira et al. (2021) with substantially longer computation times for comparable conditions. However, we agree that this is not a direct comparison of computation times. In order to avoid further misunderstanding, we have rewritten the sentence. It now reads:

> *We intentionally did not conduct explicit simulations of reactive transport in this study since our main goal here was to explore the HMC method (coupled to a flow model) in order to assess the development of mixing spots on the riparian zone and their relation to hydrological variations. We additionally showed the importance of such macroscopic mixing spots for groundwater-borne $NO_3^-$ turnover by comparing the quantitative mixing results of the HMC method with previous biogeochemical assessments carried out in the study area. For a direct quantification of nitrate removal rates, however, the use of reactive-transport models or other data-drive analyses would be indispensable. Such simulations would have allowed a comparison of observed and simulated concentration values and their dynamics for a more rigorous evaluation of model performance (Nogueira et al., 2021b). However, the additional computational effort to numerically solve the transport equations would likely also increase computational costs.*

References

*Brunner, P., Simmons, C.T.: HydroGeoSphere: A Fully Integrated, Physically Based Hydrological Model. Ground Water 50, 170–176. https://doi.org/10.1111/j.1745-6584.2011.00882.x, 2011.*

*Jones JP, Sudicky EA, McLaren RG.: Application of a fully-integrated surface–subsurface flow model at the watershed-scale: a case study. Water Resour Res., 44:W03407, DOI:10.1029/WR005603, 2008*

Nogueira, G. E. H., Schmidt, C., Brunner, P., Graeber, D. and Fleckenstein, J. H.: Transit-time and temperature control the spatial patterns of aerobic respiration and denitrification in the riparian zone, Water Resour. Res. 57, https://doi.org/10.1029/2021WR030117, 2021.

Park, Y.J., E.A. Sudicky, R.G. McLaren, and J.F. Sykes: Analysis of hydraulic and tracer response tests within mod- erately fractured rock based on a transition probability geo-statistical approach. Water Resources Research 40:W12404. DOI: 10.1029/2004WR003188, 2004.

Partington, D., Brunner, P., Simmons, C.T., Therrien, R., Werner, A.D., Dandy, G.C., Maier, H.R.: A hydraulic mixing-cell method to quantify the groundwater component of streamflow within spatially distributed fully integrated surface water-groundwater flow models. Environ. Model. Softw. 26, 886–898. https://doi.org/10.1016/j.envsoft.2011.02.007, 2011.

Schmadel, N.M., Ward, A.S., Kurz, M.J., Fleckenstein, J.H., Zarnetske, J.P., Hannah, D.M., Blume, T., Vieweg, M., Blaen, P.J., Schmidt, C., Knapp, J.L.A., Klaar, M.J., Romeijn, P., Datry, T., Keller, T., Folegot, S., Arricibita, A.I.M., Krause, S., War, A.S., Kurz, M.J., Fleckenstein, J.H., Zarnetske, J.P., Hannah, D.M., Blume, T., Vieweg, M., Blaen, P.J., Schmidt, C., Knapp, J.L.A., Klaar, M.J., Romeijn, P., Datry, T., Keller, T., Folegot, S., Marruedo Arricibita, A.I., Krause, S.: Stream solute tracer timescales changing with discharge and reach length confound process interpretation. Water Resour. Res. 52, 3227–3245. https://doi.org/10.1002/2015WR018062, 2016.

Therrien, R., and E.A. Sudicky: Three-dimensional analysis of variably-saturated flow and solute transport in discretely-fractured porous media. Journal of Contaminant Hydrology 23, no. 1–2: 1–44, 1996.

Unger, A.J.A., Q. Li, E.A. Sudicky, D. Kassenaar, E.J. Wexler, and S. Shikaze: Simulating the multi-seasonal response of a large-scale watershed with a 3D physically- based hydrologic model. Journal of Hydrology 357, no. 3–4: 317–336, 2008.

[Figure]

**Fig.S2:** Spatial representation of a perfect mixing ($d_p$) and of and arbitrary mixing ($d$) for the cases of three **(a)** and two **(b)** end-members mixing. The final mixing $d$ can be calculated as the Euclidean distance between points $d_p$ and $d$. For a three end-members mixing (3D case), any combination of fractions can be represented as a point $d$ in a 3D coordinate space, in which the maximum distance is a radius of a circle (centred at [1/3, 1/3, 1/3]) escribed on an equilateral triangle (side length of $\sqrt{2}$). Thus, the maximum distance between $d_p$ and $d$ is ($\sqrt{2} \times \sqrt{3}/3$) For a two end-members mixing (2D case), the maximum segment is the diameter of a circle (centred at [0.5, 0.5]), whereas the maximum distance between dp and d is ($\sqrt{2}/2$). The long-dashed lines in (a) delimit the solution space for any possible mixing $d$ where fractions sum up to 1. In (b) final mixing $d$ values would fall over the solid line passing through $d_p$. Example of theoretical mixings between three **(c)** and two **(d)** end-members coloured according to computed $d$ values (warmer colours indicate a more homogenous mixing); $d_p$ is indicated as a black circle. The theoretical mixings were generated with 10000 random combinations of HMC fractions that sum up to 1. For a four (or more) end-members mixing a spatial representation is not possible but the general Eq.4 would equally work.

---

## Author Comment (AC2)

The work presented here by Nogueira et al concerns the evaluation of the reactivity of the hyporheic zone, and more globally of a small stretch of a river-aquifer interface that is in a losing condition. The paper is well written and proposes a new framework for assessing the potentiel reactivity of this interface by coupling a physically based model, that simulates stream water (SW) and groundwater (GW), with a Hydraulic Mixing Cell (HMC) module. However it suffers from various flaws that must be addressed before further publication.

We are grateful for the reviewer's constructive comments, which have helped to clarify unclear points in the paper and to generally improve the manuscript. We also appreciate the suggestions for additional points and references in the discussion of our work, which we believe have enriched not only this section, but also the overall quality of the manuscript. Below are the detailed responses to the reviewer's comments.

1. The system at hand must be detailed and a significant effort must be done for positioning it in a much broader picture (see for instance the review of Flipo et al. 2014). From the title and the abstract, it must be clear that the focus of the study is a 900m stretch of a river connected to a small portion of porous medium (the lateral extent of the model seems too narrow to speak about an aquifer). The river is in a losing condition, which is not the most probable configuration as far as SW-GW are concerned since rivers constitute the water outlet for GW at the catchment scale. Finally the connection of the small portion of river stretch with a broader regional aquifer system must be explained.

   We recognize that some points may not have been clear at the beginning of the abstract and in the introduction of the manuscript. We have revised those sections according to the suggestions of the reviewer. However, we would like to additionally emphasize the following points:

   We disagree with the notion of the reviewer that the narrow lateral extent of the water bearing alluvial sediments around the stream does not justify the use of the term aquifer. The term aquifer is primarily defined in terms of a water bearing subsurface layer's permeability and transmissivity for water (to distinguish it from an aquitard or aquiclude) and not in terms of its spatial extent. The hydraulic conductivities in the gravelly and sandy formation at the site with a thickness of up to 8 m are on the order of $10^{-4}$ to $10^{-3}$ m/s (see Nogueira et al. 2021, Trauth et al., 2018; Zhang et al., 2021), which is highly permeable and justifies to call the formation an aquifer. We refer to Lohman et al. 1972 for a definition (*Definitions of selected ground-water terms – revisions and conceptual refinements*, USGS Water Supply Paper, 1988). Borelog and geophysical data indicate low-permeability clayey and silty deposits (forming an aquitard) on top of a relatively shallow, Mesozoic, vertically tilted bedrock (inhibiting lateral groundwater flow) at the base of the coarser water bearing alluvial sediments, hydraulically disconnecting the alluvial aquifer from potential groundwater in the deeper fractured bedrock. We refer to the German geological survey for information related to the underlying bedrock, which is available online at: https://produktcenter.bgr.de/terraCatalog/OpenSearch.do?search=61ac4628-6b62-48c6-89b8-46270819f0d6&type=/Query/OpenSearch.do.

We made sure to emphasize this geological setup and the associated geological information in the revised manuscript, and have added the relevant references on the description of the study site and on the presentation of the numerical model. We also brought this point back in the discussion section as suggested by the reviewer (see answer to question 3 below). It now reads (section 2.1):

> *The alluvial aquifer consists of up to 8 m-thick alluvial sediments, with grain sizes ranging from medium sands to coarse gravels, underlain by less permeable clayey-silty deposits on top of the Mesozoic bedrock forming the bottom of alluvial aquifer. Borelog and geophysical data indicate that the thickness of the alluvial aquifer slightly decreases with distance from the stream (Lutz et al., 2020; Trauth et al., 2018).*

We also respectfully disagree with the statement that losing conditions are an unlikely condition in river catchments. While we agree that at regional-scale groundwater will eventually discharge to surface water bodies such as rivers, lakes or estuaries, river reaches with temporarily or even permanently losing conditions are by no means uncommon and are not only found in arid or semi-arid regions. Local losing conditions on streams may be caused by several natural (geology, topography and climate) and human-induced conditions (Brunner et al., 2009; Irvine et al., 2012; Jones et al., 2008; Liao et al., 2014; Munz et al., 2019; Poole et al., 2008; Schilling et al., 2017; Su et al., 2007; Treese et al., 2009; Vogt et al., 2010a, 2010b). Our field site is just downstream of the transition between the steeper, mountainous upper- and the flatter, alluvial lower catchment of the Selke river. This location is on the lee side of the Harz Mountains, which block parts of the westerly winds that deliver most precipitation in Central Europe. Therefore annual precipitation is relatively low (~500 mm) and in turn groundwater recharge rates are small (on average < 100 mm) facilitating the disconnection between the alluvial aquifer and the stream. Such disconnections are a rather common condition on the lee side of mountains in larger river catchments, even in temperate and more humid climate regions. Furthermore projections indicate that losing conditions are becoming more common due to global change and increasing groundwater withdraws within alluvial aquifers (Jasechko et al., 2021).

Alongside the HMC application, with our study we also aim to emphasize the importance of losing streams in the role of riparian biogeochemical processes, especially on mixing-dependent reactions. Previous studies have already demonstrated the role of losing streams in providing DOC and other solutes to trigger and boost riparian biogeochemical turnover processes (Hester et al., 2013, 2014, 2017, 2019; Lutz et al., 2020; Munz et al., 2019; Trauth et al., 2014, 2018; Trauth and Fleckenstein, 2017), which is especially important for the turnover of groundwater-borne solutes, such as the case of Nitrate in the studied area (Gassen et al., 2017; Lutz et al., 2020; Trauth and Fleckenstein, 2017). As suggested by the reviewer, we have further emphasized this point in the abstract and in the description of the study area.

We agree with the reviewer that the connections of our findings to broader scale processes could be outlined more explicitly in our manuscript. We now indicate the connections and the relevance of our study for larger scales at different points in the text (see for instance answer to question 3 below). However, at the same time we want to be careful not to make statements at scales that are clearly beyond the scale and data of this study.

2. The main conclusions highlighted in the abstract only make sense if it is clearly stated beforehand that the stretch of river is in a losing configuration otherwise readers could be misled at the reading of the abstract. On the one hand, the highlighted results of evaluated water mixing values should be moderated in the abstract considering the remark 3. On the other hand, it seems to me that an important result of the study is not sufficiently reported in the abstract, it is the fact that the potential hot spots of reactivity of such a system in terms of nitrate removal is located at the fringe of the HZ and not directly below the leaking river.

We have emphasized the net losing conditions of the local reach in the abstract. We have also rewritten the main findings of our study in the abstract taking the suggestions into account (referring the computed percentages in terms of the model domain rather than the riparian aquifer). We agree with the reviewer that the original statements were not sufficiently highlighting the important key finding on the location of the potential hot-spots of reactivity in terms of distance from the stream channel. The respective text in the abstract now reads:

> *Our results show that on average about 50% of the water in the alluvial aquifer consists of infiltrating SW. Within about 200m around the stream the aquifer is almost entirely made up of infiltrated SW with practically no significant amounts of other water sources mixed in. On average, about 9% of the model domain could be characterized as "mixing hot-spots" (locations with more balanced fractions of the different water sources), which were mainly located at the fringe of the geochemical hyporheic zone rather than below or in the immediate vicinity of the streambed.*

3. The GW model set up must be detailed. What is the extent, in the x, y and z directions ? what are the lateral boundary conditions and also at the bottom of the system, as well as for the upstream part of the simulated porous media. If no water flux conditions are used for the lateral and the bottom of the porous medium system, it has consequences on the presented result, entailing them with a large uncertainty related to the misconception of the connection of the system to the larger regional aquifer system. A discussion on the consequence of the model set-up should be added to the paper.

We have added respective information on model setup and its extent. It now reads:

> *The simulated domain (900 × 770 × 10 m) was divided into four main hydrogeological units according to geophysical and borelog data, which further indicates the thinning of the alluvial aquifer with distance from the stream (Lutz et al., 2020; Trauth et al., 2018). Thus, the simulated domain covers most of the mapped alluvial aquifer present in the area. The bottom of the numerical model was set as a no-flow boundary in line with the less permeable clayey-silty deposits and the low-permeability bedrock at the base of the coarser alluvial sediments. The boundary conditions (BCs) on the model surface domain were defined as (i) groundwater recharge (as a fraction of daily precipitation) at the model top, (ii) specified water flux at the model stream inlet according to discharge values measured at a gauge station about 3000m upstream of the study site, and (iii) a critical depth BC at the model stream outlet (Fig.3a). The BCs on the subsurface model domain were defined as (iv) specified water flux representing ambient groundwater flow at the upstream side of the model, and (v) prescribed time-varying hydraulic heads at the downstream side of the model (Fig.3a). The other lateral subsurface boundaries of the model domain were set as no-flow boundaries based on field observations indicating that GW flowlines are somewhat parallel to the stream with distance.*

For more details on the model setup we refer the readers to Nogueira et al. (2021).

We have also added a discussion on the consequence of the model setup to the larger-regional aquifer system. We agree with the reviewer that this point was not discussed in the text. We highly appreciate this suggestion from the reviewer. It now reads (section 4.5):

> *Even though the numerical model matched field observations well, it represents a simplification of reality (a characteristic inherent to all models), which in turn results in some limitations and uncertainties. For instance, based on available geophysical data we have assumed the clayey-silty formation on top of the vertically tillted low-permeability bedrock as the bottom of the alluvial aquifer and impermeable in the model. We assumed that the alluvial aquifer has a limited lateral extent (Lutz et al., 2020; Trauth et al., 2018), which was backed by geophysical data and the presence of bedrock outcrops along parts of the lateral model boundaries. These assumptions and the chosen model geometry, however, may not fully account for larger-scale hydrological fluxes, which are inherent to nested SW-GW systems. For instance, as showed by Flipo et al. (2014) and by other studies (Boulton et al., 1998; Magliozi et al., 2018; Toth 1963), SW-GW system are connected interfaces, which are linked to each other through different spatio-temporal processes. For instance, longer and deeper flowpaths that might have been not represented in our numerical model could lead to the development of additional mixing spots at greater depths or distances from the stream (Lessels et al., 2016). This could further emphasize and explain how alluvial aquifers and riparian zones act as buffer zones connecting low-frequency processes occurring at regional scale and high-frequency processes occurring in the stream network (Ebeling et al., 2021; Flipo et al., 2014; Rivett et al., 2008; Sun et al. 2017). Equally, lateral influx of groundwater through the lateral boundaries of the model domain could also effect the dynamics and main directions of GW flow paths and therefore SW-GW mixing spots development. However, head data at the site did not show any indications of such effects. Furthermore the specific geology of the site with shallow, low-permeability mesozoic bedrock strata, which inhibit lateral groundwater movement as they are vertically tilted, rules out the presence of a laterally extensive, continuous regional aquifer. Exchange fluxes between the shallow alluvial aquifer with deeper groundwater were therefore considered to be negligible.*

4. One way to clarify the paper is to add a summary of the other Nogueira et al papers

We appreciate this suggestion. We have added additional information from our earlier paper throughout the text taking into account the comments from the reviewers. For more details, the readers are directly referred to Nogueira et al 2020 and 2021.

5. The added value of using HMC rather than a fully coupled transport model is not clear and is in the current state of the paper an affirmation, not a scientific statement. As it is stated that Nogueira et al in press used the transport module of HydroGeoSphere, a comparative assessment of computational duration should be provided. This quantification is essential because from line 609-618 it seems more efficient to directly use a transport model than a HMC for the quantitative assessment of the stream-aquifer interface in terms of nitrates removal.

We agree that the HMC framework presented here does not allow for a direct quantification of nitrate removal rates. It is rather a complementary tool that indicates locations where the potential for removal can be high due to SW-GW mixing. For a direct quantification, reactive-transport models or additional field data in combination with data-driven analyses would be needed. However, the parameterization of such a transport model requires significant amounts of spatially distributed data (e.g. local concentrations) to constrain parameter ranges (e.g. reaction rate coefficients). Instead we decided to use the field data from Gassen et al. (2017) for a more qualitative evaluation of our simulated patterns of mixing potentials obtained from

the robust HMC model, assuming that reactions facilitated by the mixing of the different water sources can explain the sharp concentration fronts observed by Gassen et al. (an assumption also implicitly made by the authors of that study). In that regard results from the HMC method, which can be well constrained with the existing hydraulic data and does not require extensive data on transport and reaction parameters, can serve as a proxy and complementary tool to interpret observed concentration patterns. We have emphasized these points in the discussion section 4.5.

We would like to further clarify that Nogueira et al. (2021) did not use a transport routine within HGS. Instead they developed a sequentially coupled reactive-transport model based on the flow simulations from the HGS and other field data since HGS does not allow for temperature-dependent reaction rates to be implemented at the moment. Therefore a comparison of computation-times between this model and the model in this study would not be meaningful. To clarify this we have rewritten the text in the respective section of the manuscript. It now reads:

> *We intentionally did not conduct explicit simulations of reactive transport in this study since our main goal here was to explore the HMC method (coupled to a flow model) to assess the development of mixing spots on the riparian zone and their relation to hydrological variations. Spatial patterns of mixing hot-spots can provide a meaningful proxy for the interpretation of reactivity patterns in the absence of extensive data for the parameterization of an explicit reactive transport model. Along those lines we could illustrate the importance of such macroscopic mixing spots for groundwater-borne $NO_3^-$ turnover by comparing the quantitative mixing results of the HMC method with previous biogeochemical assessments carried out in the study area. For a direct quantification of nitrate removal rates, however, the use of reactive-transport models or additional field data combined with data-drive analyses would be needed. Such simulations would have allowed a comparison of observed and simulated concentration values and their dynamics for a more rigorous evaluation of model performance (Nogueira et al., 2021b).*

6. Errors in mathematical formulas are unacceptable and must be corrected :
   1. Eq 1. f_(w)^ t-1 not defined, as well as vbc_k^t

      Thank you for pointing this out. $Vbc_k^t$ should be $Vbc_w^t$ in the equation as it is in the text (we have replaced $\underline{k}$ for $\underline{w}$ in the equation). On the other hand, we believe that the terms $f_{i(w)}^{(t-1)}$ and $f_{j(w)}^{(t-1)}$ have been defined in the text just after the equation is presented.

   2. Eq 2 not homogeneous in terms of units between left hand side and right hand side

      Thank you for bringing this to our attention. There was a missing term on the equation and in the explanatory text. We have corrected the problem. It now reads:

      $$V_w = \frac{\sum_{p=1}^{P} (V_p \, f_{w,p})}{V_{tot}} \times 100\%$$

      In line with the suggestion from the reviewer to clarify the *Integration* function (see answer to other remarks 5), we have also added the following sentence before the Eq.2:

*The function integrates the numerical cells within the simulated domain taking into account only the fraction of interest that comprises each cell volume. The calculation sums the resulting quantities over the domain to produce the integrated result, which is then normalized by the total volume of the simulated domain ($V_{tot}$). Thus, the resulting volume represents a percentage of the total simulated domain*

3.  Eq 4 the denominator seems wrong, please check and either add the original reference or detail the math. L 250 the value of the denominator of eq 4 in case only two pools of water are concerned is 1, root square(2)/2 as stated by the authors.

    Based on comments from the other reviewer, we have now added more explanation on the development of the equation and on how the mixing degree can be calculated based on an analytical geometry approach. We have added a figure to the supplementary material (see now Fig.S2 – also attached at the end of this document) to illustrate the concept.

    In brief, for a three end-member mixing case, any combination of three different source water fractions can be represented as a point *d* in a 3D coordinate space, in which the maximum Euclidian distance between point d and the point of equal mixing (equal fractions of all mixing members) within the mixing space is the radius of a circle (centred at [1/3, 1/3, 1/3]) escribed on an equilateral triangle (side length of $\sqrt{2}$). For a two end-member mixing case, the maximum segment is the diameter of a circle (centred at [0.5, 0.5]) with side length of $\sqrt{2}/2$. For a four (or more) end-member mixing case a spatial representation is not possible but equation (4) would equally apply. Therefore we would like to keep it in the text.

4.  Same problem in eqs 5 and 6

    Equally to Eq.2, there were missing terms on the equations and in the explanatory text. They now read:

$$V_d = \frac{\sum_{p=1}^{P} (V_p\, d)}{V_{tot}} \times 100\%$$

$$V_{d\_HZ} = \frac{\sum_{p=1}^{P} (V_p\, d)}{V_{HZ}} \times 100\%$$

7.  The discussion about the reactivity of the interface should be enriched with other important references such as Newcomer et al. 2018, especially providing arguments on the added value of a 3D approach.

    We have rewritten this discussion following the suggestion from the reviewer on the added value of our 3D approach. We recognize that this has improved this section. It now reads (section 4.3):

*Previous work on hyporheic reactivity has often been carried using 1D or 2D model setups focusing on biogeochemical processes in direct vicinity of the streambed (Hester et al., 2014, 2019; Newcomer et al., 2018). This study, using a larger-scale 3D model also considers lateral SW-GW exchange fluxes over longer distances into the riparian aquifer the associated longer-term mixing processes further away from the stream channel. In line with results from Nogueira et al. (2021b) and Trauth et al. (2018), results from our 3D model coupled with the HMC method reinforce that such larger-scale and long-term processes are important around losing streams for the creation of mixing hot-spots at larger distance from the stream. These mixing hot spots can facilitate mixing-dependent biogeochemical reactions, which may significantly contribute to the net turnover of groundwater-borne solutes at the stream corridor scale. These processes may have been overseen in small-scale studies, which have focused on the immediate interface between the stream channel and the alluvial groundwater only.*

Nevertheless, we would like to emphasize that Newcomer et al., 2018 did not consider groundwater-borne solutes such as Nitrate ("*we simplify our model to the scenario where groundwater $NO_3^-$ contamination is not present*"), and thus could not evaluate the links between SW-GW mixing and mixing-triggered reactions, which was the focus of our study. In turn, they have only considered stream-borne solutes (e.g., DOC, $NO_3^-$) and their turnover in the hyporheic zone below the stream bed. We also believe that the Newcomer et al. study, although it enriches our discussion, is less in line with our study than others studies, which have explicitly included groundwater-borne Nitrate in their simulations, for instance the studies by Hester et al. (2017, 2019) and Trauth et al. (2014).

8. Sec 2.4.1 the authors mention that the origin of water from the flood plain can be neglected, then developing eq 4 in that specific case. It is confusing since they use 3 origins in the remaining of the paper. Section 2.4.1 must be reworked l235-271

We recognize this section was confusing and we have rewritten it in order to clarify the idea behind the reduction from a three to a two end-member mixing model. We still think, however, that it is important to keep the three end-members represented throughout the manuscript since there are time periods when the three components are all present in the saturated portion of the domain in high fractions. These episodes are an important characteristic of the temporal exchange dynamics of the coupled GW-SW system. Therefore we would like to keep the three end-member case in the manuscript. We hope that with the additional explanation this idea is now clear in this section. Following Eq. 4, it now reads:

*where $f_1$, $f_2$, and $f_w$ represent HMC fractions. Based on preliminary results, we have observed that actual volumes of $f_{FW}$ were very low in comparison to $f_{GW}$ and $f_{SW}$ in the fully saturated portion of the domain as it will be demonstrated in section 3.2. This occurs because recharge from precipitation is very low at the site (Nogueira et al., 2021b), and the percolation of water from the top of the model domain is limited to occasional episodes. Therefore, we have employed a simplified version of the Eq.4 considering a two end-member mixing only. To do so, we combined the two end-members $f_{GW}$ and $f_{FW}$ to a single one (e.g., [$f_{GW}+ f_{FW}$], Fig.S2, supplementary material), which reduces the mixing model to a two 2D case. This streamlined two end-member mixing is the preferred one used throughout the manuscript because otherwise resulting d values would be consistently very low in the simulations, which would impair their further analyses.*

Other remarks

1. L. 127 Please write the explanation of Fig. 2 in a paragraph at the beginning of section 2 Method. It is not currently detailed, only the Figure is in the document.

   We have added a sentence to briefly explain the figure in the text. It now reads:

   > *In brief, following field data collection, a 3D numerical flow model was developed and calibrated against the collected field data (Nogueira et al., 2021b). The HMC method is then coupled to the numerical model, whereas results are additionally evaluated according to additional hydrochemical data (i.e., water samples) for further mapping of water sources and analysis of mixing degrees within the riparian zone. In the subsequent sections we detail each step and the methods used.*

2. L. 141 AT each time step

   We have corrected it.

3. Fig 3a. Scales are not readable, especially in the Z direction. Overall the readability of the whole figure must be improved. The reader should be able to read the piezometer names

   We apologize for that. We made sure to increase the legend size, as well as the names of the piezometers to guarantee their good readability. We also increased the indication of the elevation isolines and changed their colors slightly to improve their readability.

4. L 180 grammar issue

   We have corrected it.

5. L 196 what is the integration function of Tecplot, please explain the math instead

   We have added the following sentence in order to clarify the utilized function just before the Eq.2:

   > *The function integrates the numerical cells within the simulated domain taking into account only the fraction of interest that comprises each cell volume. The calculation sums the resulting quantities over the domain to produce the integrated result, which is then normalized by the total volume of the simulated domain ($V_{tot}$). Thus, the resulting volume represents a percentage of the total simulated domain*

6. L 204 50% OF stream water

   We have corrected it.

7. L210 WHILE most

   We have corrected it.

8. Fig 6 and 7 are too small and therefore not very informative. The authors must select more dedicated illustrations that correspond more closely to their message in the text

   In our opinion, the figures convey key information on the spatial distribution and variations (in terms of expected minimum, maximum and average distribution) of the different water

fractions in the simulated domain, as well as of the mixing degrees. In order to clarify this point and their relevance, we have added an extra sentence before the figure. It now reads:

*The plots indicate the minimum and maximum possible distributions of each water fraction in the domain, as well as their typical distribution throughout the simulation period.*

We have done the same for Figure 7. Moreover, we have enlarged the legends on the figures to improve their readability. We hope this has solved the issue with the figures. As we think they provide key information linked to the mapping of different water fractions and mixing degrees in the simulated domain, we would like to keep them in the manuscript.

References

Brunner, P., Simmons, C. T. and Cook, P. G.: Spatial and temporal aspects of the transition from connection to disconnection between rivers, lakes and groundwater, J. Hydrol., 376(1–2), 159–169, doi:10.1016/j.jhydrol.2009.07.023, 2009.

Ebeling, P., Dupas, R., Abbott, B., Kumar, R., Ehrhardt, S., Fleckenstein, J.H., Musolff, A.: Long-Term Nitrate Trajectories Vary by Season in Western European Catchments. Global Biogeochem. Cycles 35, 1–19. https://doi.org/10.1029/2021GB007050, 2021.

Gassen, N., Griebler, C., Werban, U., Trauth, N. and Stumpp, C.: High Resolution Monitoring Above and Below the Groundwater Table Uncovers Small-Scale Hydrochemical Gradients, Environ. Sci. Technol., 51, 9, doi:10.1021/acs.est.7b03087, 2017.

Hester, E. T., Young, K. I. and Widdowson, M. A.: Mixing of surface and groundwater induced by riverbed dunes: Implications for hyporheic zone definitions and pollutant reactions, Water Resour. Res., 49(9), 5221–5237, doi:10.1002/wrcr.20399, 2013.

Hester, E. T., Young, K. I. and Widdowson, M. A.: Controls on mixing-dependent denitrification in hyporheic zones induced by riverbed dunes: A steady state modeling study, Water Resour. Res., 50(11), 9048–9066, doi:10.1002/2014WR015424, 2014.

Hester, E. T., Cardenas, M. B., Haggerty, R. and Apte, S. V.: The importance and challenge of hyporheic mixing, Water Resour. Res., 53(5), 3565–3575, doi:10.1002/2016WR020005, 2017.

Hester, E. T., Eastes, L. A. and Widdowson, M. A.: Effect of Surface Water Stage Fluctuation on Mixing-Dependent Hyporheic Denitrification in Riverbed Dunes, Water Resour. Res., 55(6), 4668–4687, doi:10.1029/2018WR024198, 2019.

Irvine, D. J., Brunner, P., Franssen, H. J. H. and Simmons, C. T.: Heterogeneous or homogeneous? Implications of simplifying heterogeneous streambeds in models of losing streams, J. Hydrol., 424–425, 16–23, doi:10.1016/j.jhydrol.2011.11.051, 2012.

Jasechko, S., Seybold, H., Perrone, D., Fan, Y. and Kirchner, J. W.: Widespread potential loss of

*streamflow into underlying aquifers across the USA, Nature, 591(7850), 391–395, doi:10.1038/s41586-021-03311-x, 2021.*

*Jones, K. L., Poole, G. C., Woessner, W. W., Vitale, M. V., Boer, B. R., O'Daniel, S. J., Thomas, S. A. and Geffen, B. A.: Geomorphology, hydrology, and aquatic vegetation drive seasonal hyporheic flow patterns across a gravel-dominated floodplain, Hydrol. Process., 22(13), 2105–2113, doi:10.1002/hyp.6810, 2008.*

*Liao, Z., Osenbrück, K. and Cirpka, O. A.: Non-stationary nonparametric inference of river-to-groundwater travel-time distributions, J. Hydrol., 519(PD), 3386–3399, doi:10.1016/j.jhydrol.2014.09.084, 2014.*

*Lutz, S. R., Trauth, N., Musolff, A., Van Breukelen, B. M., Knöller, K. and Fleckenstein, J. H.: How Important is Denitrification in Riparian Zones? Combining End-Member Mixing and Isotope Modeling to Quantify Nitrate Removal from Riparian Groundwater, Water Resour. Res., 56(1), doi:10.1029/2019WR025528, 2020.*

*Munz, M., Oswald, S. E., Schäfferling, R. and Lensing, H.-J.: Temperature-dependent redox zonation, nitrate removal and attenuation of organic micropollutants during bank filtration, Water Res., 162(10), 225–235, doi:10.1016/j.watres.2019.06.041, 2019.*

*Newcomer, M. E., Hubbard, S. S., Fleckenstein, J. H., Maier, U., Schmidt, C., Thullner, M., Ulrich, C., Flipo, N. and Rubin, Y.: Influence of Hydrological Perturbations and Riverbed Sediment Characteristics on Hyporheic Zone Respiration of CO2 and N2, J. Geophys. Res. Biogeosciences, 123(3), 902–922, doi:10.1002/2017JG004090, 2018.*

*Nogueira, G. E. H., Schmidt, C., Brunner, P., Graeber, D. and Fleckenstein, J. H.: Transit-time and temperature control the spatial patterns of aerobic respiration and denitrification in the riparian zone, Water Resour. Res., doi:10.1029/2021WR030117, 2021.*

*Poole, G. C., O'Daniel, S. J., Jones, K. L., Woessner, W. W., Bernhardt, E. S., Helton, A. M., Stanford, J. A., Boer, B. R. and Beechie, T. J.: Hydrologic spiralling: the role of multiple interactive flow paths in stream ecosystems, River Res. Appl., 24(7), 1018–1031, doi:10.1002/rra.1099, 2008.*

*Schilling, O. S., Gerber, C., Partington, D. J., Purtschert, R., Brennwald, M. S., Kipfer, R., Hunkeler, D. and Brunner, P.: Advancing Physically-Based Flow Simulations of Alluvial Systems Through Atmospheric Noble Gases and the Novel37Ar Tracer Method, Water Resour. Res., 53(12), 10465–10490, doi:10.1002/2017WR020754, 2017.*

*Su, G. W., Jasperse, J., Seymour, D., Constantz, J. and Zhou, Q.: Analysis of pumping-induced unsaturated regions beneath a perennial river, Water Resour. Res., 43(8), 1–14, doi:10.1029/2006WR005389, 2007.*

*Trauth, N. and Fleckenstein, J. H.: Single discharge events increase reactive efficiency of the hyporheic zone, Water Resour. Res., 53(Jan), 779–798, doi:10.1111/j.1752-1688.1969.tb04897.x, 2017.*

*Trauth, N., Schmidt, C., Vieweg, M., Maier, U. and Fleckenstein, J. H.: Hyporheic transport and biogeochemical reactions in pool-riffle systems under varying ambient groundwater flow conditions, J. Geophys. Res. Biogeosciences, 119(5), 910–928, doi:10.1002/2013JG002586, 2014.*

Trauth, N., Musolff, A., Knöller, K., Kaden, U. S., Keller, T., Werban, U. and Fleckenstein, J. H.: River water infiltration enhances denitrification efficiency in riparian groundwater, Water Res., 130, 185–199, doi:10.1016/j.watres.2017.11.058, 2018.

Treese, S., Meixner, T. and Hogan, J. F.: Clogging of an Effluent Dominated Semiarid River: A Conceptual Model of Stream-Aquifer Interactions, JAWRA J. Am. Water Resour. Assoc., 45(4), 1047–1062, doi:10.1111/j.1752-1688.2009.00346.x, 2009.

Vogt, T., Schneider, P., Hahn-Woernle, L. and Cirpka, O. A.: Estimation of seepage rates in a losing stream by means of fiber-optic high-resolution vertical temperature profiling, J. Hydrol., 380(1–2), 154–164, doi:10.1016/j.jhydrol.2009.10.033, 2010a.

Vogt, T., Hoehn, E., Schneider, P., Freund, A., Schirmer, M. and Cirpka, O. A.: Fluctuations of electrical conductivity as a natural tracer for bank filtration in a losing stream, Adv. Water Resour., 33(11), 1296–1308, doi:10.1016/j.advwatres.2010.02.007, 2010b.

Winter, C., Lutz, S.R., Musolff, A., Kumar, R., Weber, M., Fleckenstein, J.H.: Disentangling the Impact of Catchment Heterogeneity on Nitrate Export Dynamics From Event to Long-Term Time Scales. Water Resour. Res. 57, 1–24. https://doi.org/10.1029/2020WR027992, 2021.

Zhang, Z. Y., Schmidt, C., Nixdorf, E., Kuang, X. and Fleckenstein, J. H.: Effects of Heterogeneous Stream-Groundwater Exchange on the Source Composition of Stream Discharge and Solute Load, Water Resour. Res., 57(8), 1–19, doi:10.1029/2020WR029079, 2021.

[Figure]

**Fig.S2:** Spatial representation of a perfect mixing ($d_p$) and of and arbitrary mixing ($d$) for the cases of three **(a)** and two **(b)** end-members mixing. The final mixing $d$ can be calculated as the Euclidean distance between points $d_p$ and $d$. For a three end-members mixing (3D case), any combination of fractions can be represented as a point $d$ in a 3D coordinate space, in which the maximum distance is a radius of a circle (centred at [1/3, 1/3, 1/3]) escribed on an equilateral triangle (side length of $\sqrt{2}$). Thus, the maximum distance between $d_p$ and $d$ is ($\sqrt{2} \times \sqrt{3}/_3$) For a two end-members mixing (2D case), the maximum segment is the diameter of a circle (centred at [0.5, 0.5]), whereas the maximum distance between dp and d is ($\sqrt{2}/2$). The long-dashed lines in (a) delimit the solution space for any possible mixing $d$ where fractions sum up to 1. In (b) final mixing $d$ values would fall over the solid line passing through $d_p$. Example of theoretical mixings between three **(c)** and two **(d)** end-members coloured according to computed $d$ values (warmer colours indicate a more homogenous mixing); $d_p$ is indicated as a black circle. The theoretical mixings were generated with 10000 random combinations of HMC fractions that sum up to 1. For a four (or more) end-members mixing a spatial representation is not possible but the general Eq.4 would equally work.